# HMoRA: Making LLMs More Effective with Hierarchical Mixture of LoRA Experts

**Mengqi Liao, Wei Chen, Junfeng Shen, Shengnan Guo & Huaiyu Wan**[*]
Department of Computer Science
Beijing Jiaotong University
{mqliao,w_chen,jfshen,guoshn,hywan}@bjtu.edu.cn

## Abstract

Recent studies have combined Mixture of Experts (MoE) and Parameter-Efficient Fine-tuning (PEFT) to fine-tune large language models (LLMs), holding excellent performance in multi-task scenarios while remaining resource-efficient. However, existing MoE approaches still exhibit the following limitations: (1) Current methods fail to consider that different LLM layers capture features at varying levels of granularity, leading to suboptimal performance. (2) Task-level routing methods lack generalizability to unseen tasks. (3) The uncertainty introduced by load imbalance loss undermines the effective specialization of the experts. To address these challenges, we propose HMoRA, a **H**ierarchical fine-tuning method that combines **Mo**E and Lo**RA**, employing hybrid routing that integrates token-level and task-level routing in a hierarchical manner. This hierarchical hybrid routing allows the model to more efficiently capture both fine-grained token information and broader task contexts. To improve the certainty of expert selection, a novel routing auxiliary loss is introduced. This auxiliary function also enhances the task router's ability to differentiate tasks and its generalization to unseen tasks. Additionally, several optional lightweight designs have been proposed to significantly reduce both the number of trainable parameters and computational costs. Experimental results demonstrate that HMoRA outperforms full fine-tuning across multiple NLP benchmarks, while fine-tuning only 3.9% of the parameters. The code is available on: `https://github.com/LiaoMengqi/HMoRA`.

## 1 Introduction

Large language models (LLMs) have made significant strides and achieved impressive capabilities in various natural language processing (NLP) (Liu et al., 2023; Touvron et al., 2023; Team et al., 2024) tasks, such as machine translation, text generation, and question answering. However, with models reaching tens or hundreds of billions of parameters, the computational and memory costs for training have increased substantially (Kaplan et al., 2020). Therefore, reducing these costs while maintaining performance has become a critical challenge.

In recent years, Mixture of Experts (MoE) (Jacobs et al., 1991; Cai et al., 2024) has attracted growing interest in the research community. The MoE is a neural network architecture that activates a subset of expert modules for each input, utilizing a routing mechanism to determine which experts are engaged. MoE models enable the expansion of model capacity without significantly increasing computational costs. Recent studies have successfully integrated MoE into LLMs, significantly improving scalability and efficiency (Lepikhin et al., 2020; Du et al., 2022; Jiang et al., 2024), enhancing generalization in multi-task learning (Fedus et al., 2022). Furthermore, Shen et al. (2023) demonstrates additional performance gains in LLMs by combining MoE with instruction fine-tuning. However, due to the presence of multiple expert modules, standard MoE models have an extremely large number of parameters, which poses challenges for storage, deployment, and practical application. Recent studies have combined Low-Rank Adaptation (LoRA) (Hu et al., 2021) with MoE for large language model fine-tuning, leveraging LoRA's parameter efficiency alongside MoE's ability to en-

---

[*]Corresponding author.

hance performance in complex and multi-task scenarios (Zadouri et al., 2023; Li et al., 2024a; Tian et al., 2025).

However, existing MoE models still face several challenges. (1) Combining MoE routing at different granularities, such as token-level and task-level, allows the model to more effectively capture information across diverse representational hierarchies (Kudugunta et al., 2021). However, existing MoE LLMs that employ multi-granular routing (Ren et al., 2023) fail to account for the fact that different layers of LLM capture features at distinct granularities (Geva et al., 2021). Consequently, the efficiency of capturing multi-granular information remains suboptimal. (2) What's more, existing task-level routing MoE methods rely on task labels, which restricts their capacity to generalize to unseen tasks (Ren et al., 2023; Feng et al., 2024; Liu et al., 2024). Some methods (Kudugunta et al., 2021; Zadouri et al., 2023) eliminate reliance on task labels by using sentence representations for routing, a technique known as sentence-level routing. However, experiments show that sentence-level routing leads to suboptimal performance. (3) Additionally, existing load balancing loss functions (Shazeer et al., 2017; Fedus et al., 2022) often result in a lack of certainty in routing results, leading to unstable routing and undermining the specialization of experts.

To address these challenges, we propose a hybrid routing method that combines task-level and token-level routing in a **h**ierarchical manner, allowing shallow layers of LLMs capture fine-grained token-level nuances, while deeper layers focus on broader task-level understanding. To tackle the issue of uncertainty in existing routing methods, a novel auxiliary function is proposed to enhance the certainty of expert selection and maintain balanced experts selection, improving expert specialization. Moreover, although our task routing is based on sentence representations, the auxiliary loss enhances the ability of task router to distinguish between different tasks in an unsupervised manner and demonstrates generalization to unseen tasks. Additionally, some lightweight designs are offered to reduce both the trainable parameters and computational costs without significantly compromising performance. Due to the high cost of training a standard MoE model from scratch, we validate the effectiveness of our proposed method in the context of fine-tuning. Specifically, we implement our approach as a PEFT method that integrates **Mo**E and Lo**RA**, which we refer to as **HMoRA**. Our contributions can be summarized as follows:

- We propose a hierarchical hybrid routing mechanism that more efficiently captures information at different granularities across various layers of LLMs.

- We introduce a novel auxiliary function that enhances the certainty of routing methods while maintaining balance in the experts selection, thereby improving expert specialization.

- By incorporating our auxiliary loss, the task router can learn to differentiate tasks in an unsupervised manner and generalize to unseen tasks.

- We provide several optional lightweight designs that further reduce both trainable parameters and computational costs.

- We train on a multi-task dataset and evaluate performance across various NLP benchmarks. With only 3.9% of the parameters compared to full fine-tuning, our method outperforms full fine-tuning on multiple benchmarks.

## 2 PREILIMINARY

Our work builds on widely adopted causal decoder LLMs (Radford et al., 2019; Touvron et al., 2023; Yang et al., 2024), functioning as a plugin integrated into the dense layers of these models. Below, we will briefly introduce the dense layers in causal decoder LLMs and the mutil-task fine-tuning.

**Dense layers in Causal Decoder LLMs.** Causal decoder-based LLMs are built on the Transformer decoder architecture. Each layer contains a self-attention mechanism and a feed-forward network (FFN). The self-attention mechanism can vary across different LLMs, such as the multi-head attention or the multi-query attention (Ainslie et al., 2023). In general, it involves three dense layers ($W_q$, $W_k$, $W_v$) computing the query, key, and value, and a fourth dense layer ($W_o$) aggregating the attention heads. FFN typically consists of two dense layers with a non-linear activation function, as described by the following equation:

$$\text{FFN}(X) = \phi(XW_{\text{up}})W_{\text{down}}. \tag{1}$$

Here, $X$ is the input to the FFN. While some LLMs may include bias terms in these dense layers, they are omitted here for simplicity. If the activation function $\phi$ is a gated activation function like SwiGLU (Shazeer, 2020), it introduces an additional dense layer, $W_{\text{gate}}$. Our work focus on these dense layers ($W_{\text{q}}, W_{\text{k}}, W_{\text{v}}, W_{\text{o}}, W_{\text{up}}, W_{\text{down}}$, and $W_{\text{gate}}$) where HMoRA will be integrated.

**Supervised Fine-Tuning.** Supervised fine-tuning is a method specifically tailored to adapt LLMs to generate outputs that align with given instructions or prompts. Multi-task fine-tuning trains the model on a variety of tasks and input formats, enabling it to acquire general problem-solving skills rather than being specialized to a single dataset (Wei et al., 2021; Chung et al., 2024). During fine-tuning, each training sample consists of an input token sequence $\mathcal{T}^{\text{in}}$ and a corresponding target token sequence $\mathcal{T}^{\text{tg}}$. The loss function for fine-tuning is defined as:

$$\mathcal{L}_{\text{LM}} = -\sum_{i=1}^{n} \log(P_{\text{LM}}(\mathcal{T}_i^{\text{tg}} \mid \mathcal{T}^{\text{in}} : \mathcal{T}_{<i}^{\text{tg}})). \tag{2}$$

Here, $n$ is the length of the target sequence $\mathcal{T}^{\text{tg}}$, $P_{\text{LM}}(\cdot)$ represents the predicted probability of the $i$-th target token $\mathcal{T}_i^{\text{tg}}$, conditioned on the input sequence $\mathcal{T}^{\text{in}}$ and all predicted tokens in the target sequence. The model is optimized to maximize the likelihood of generating the correct target sequence.

## 3 THE HMoRA METHOD

In this section, we will elaborate on HMoRA as illustrated in Figure 1. We first introduce how to combine LoRA and MoE, leveraging the strengths of both approaches (Section 3.1). Then, we present the hybrid routing mechanism which combines both token-level and task-level routing in a hierarchical manner to capture fine-grained and broader contextual information (Section 3.2). Next, we introduce a novel auxiliary loss aimed at enhancing routing certainty while maintaining a balanced selection of experts, thereby improving expert specialization. The combination of our auxiliary loss with hybrid routing enhances the task router's ability to distinguish between tasks and even generalize to unseen tasks, improving overall performance in multi-task scenarios (Section 3.3).

### 3.1 MIXTURE OF LORA EXPERTS

We insert **MoRA** blocks (mixture of LoRA expert blocks), which consist of a set of LoRA experts and a router $R$, as plugins into the dense layers of LLMs. This combination of LoRA and MoE leverages the parameter efficiency of LoRA while benefiting from the strong multi-task performance of MoE. The forward pass for each expert is defined as:

$$E_i = X W_{\text{A}_i} W_{\text{B}_i}, \tag{3}$$

where $W_{\text{A}_i} \in \mathbb{R}^{d_{\text{in}} \times r}$ and $W_{\text{B}_i} \in \mathbb{R}^{r \times d_{\text{out}}}$ are low-rank matrices, with $r \ll d_{\text{in}}, d_{\text{out}}$. $W_{\text{A}_i}$ is randomly initialized, while $W_{\text{B}_i}$ is set to zero, ensuring consistency with the pre-trained state at the start of fine-tuning. $X$ is the input to the dense layer, with a dimension of $d_{\text{in}}$, and is processed in batches. The forward pass of the dense layer, with the MoRA block inserted, is defined as:

$$Y = \sum_{i=1}^{e} g_i E_i, \tag{4}$$

$$Z = XW + Y, \tag{5}$$

where $E_i$ is the output of the $i$-th expert, $g_i$ represents the gate value computed by the router for the $i$-th expert, $e$ is the number of experts, and $W$ is the original dense layer. The final output $Z$ denotes the combination of the expert outputs $Y$ with the output of original dense layer. During training and inference of LLMs, we replace the original dense layer output with $Z$. Here, we freeze $W$ and only update the parameters of experts and router, significantly reducing the parameter count compared to full fine-tuning and standard MoE approaches.

The router $R$ can be viewed as a function $f_R(\cdot) : \mathbb{R}^{d_{\text{in}}} \to \mathbb{R}^e$ that maps the input $X$ to a distribution $g \in \mathbb{R}^e$, where $g$ represents the gate values used to select experts. In this paper, we investigate soft routing and top-$k$ routing, combined with different auxiliary losses, as the routing methods for the routers. A detailed explanation of routing methods is provided in Appendix A.

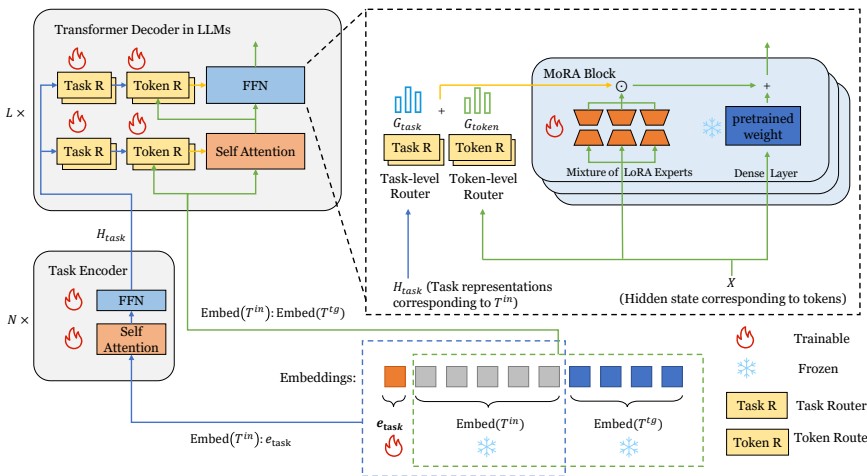

Figure 1: The HMoRA architecture combines token-level and task-level routing, utilizing a task encoder and task embedding to obtain task representations. MoRA blocks are integrated into the dense layers of LLMs.

## 3.2 HIERARCHICAL HYBRID ROUTING

Most previous MoE methods use token-level routing, which helps experts learn token-level features but fails to capture task-level information. Methods like MoELoRA (Liu et al., 2024) and MoA (Feng et al., 2024) use task-level routing but overlook fine-grained token-level details, crucial for capturing data subtleties. Although Ren et al. (2023) and Kudugunta et al. (2021) incorporate both token-level and task-level routing, they overlook the fact that shallow layers in LLMs primarily capture token-level information, while deeper layers focus more on semantic-level information (Geva et al., 2021). To address these limitations, we propose a hierarchical hybrid approach that more effectively exploits multi-granular information for routing.

To perform task-level routing, we first need to derive a task representation from the input. We define a task embedding $e_{\text{task}}$, which is concatenated with the embeddings of the input tokens $\mathcal{T}^{in}$ and processed by a task encoder, TaskEncoder($\cdot$). This process is formalized as

$$H_{\text{task}} = \text{TaskEncoder}(\text{Embed}(\mathcal{T}^{in}) : e_{\text{task}}), \tag{6}$$

where TaskEncoder($\cdot$) can be a single or multi-layer Transformer encoder. The output of the task encoder corresponding to $e_{\text{task}}$ serves as the task representation. Embed($\cdot$) is the LLM's embedding layer, with $e_{\text{task}}$ being trainable while the rest of the embeddings remain frozen. We initialize $e_{\text{task}}$ using the question mark symbol's embedding, providing a meaningful starting point for task differentiation.

To combine token-level and task-level routing, we merge a task router $R_s$ and a token router $R_t$ into a unified router. The task router computes the task-level routing results, denoted as $g_{\text{task}} = f_{R_s}(H_{\text{task}})$, which is calculated once for each input $\mathcal{T}^{in}$. The token router computes the token-level routing results, denoted as $g_{\text{token}} = f_{R_t}(X)$, which is calculated for each token in both $\mathcal{T}^{in}$ and $\mathcal{T}^{tg}$. We then combine the task-level and token-level routing results to form the final gate values for each token,

$$g = \alpha^{(l)} g_{\text{task}} + (1 - \alpha^{(l)}) g_{\text{token}}, \tag{7}$$

where $\alpha^{(l)}$ represents the proportion of the two types of routing results and $l$ indicates that the router is at Layer $l$ of the LLM. We define $\alpha^{(l)}$ as follows:

$$\alpha^{(l)} = \sigma \left( -\epsilon + 2 \times \epsilon \times \frac{l}{L} + \mu \right) \tag{8}$$

Here, $\sigma(\cdot)$ represents the sigmoid function, and $L$ is the total number of layers in the LLM. $\epsilon$ and $\mu$ are hyperparameters that flexibly control the variation of $\alpha^{(l)}$ as $l$ changes. This allows different

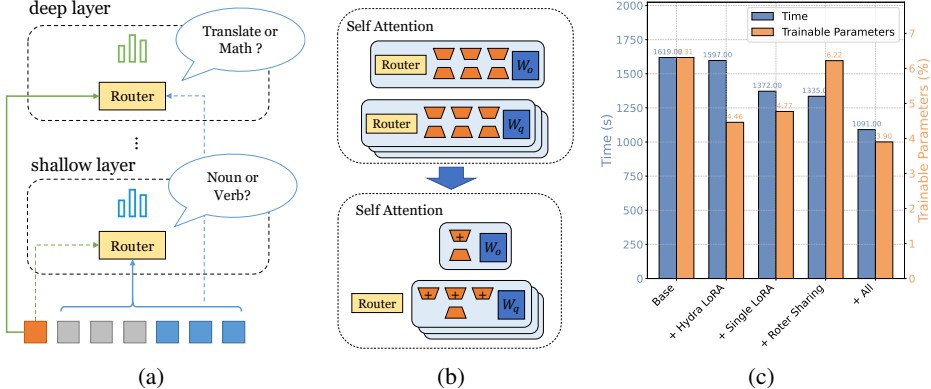

Figure 2: (a) More hierarchical LLMs by applying hierarchical hybrid routing, where shallow layers focus on fine-grained token-level distinctions and deeper layers shift towards broader task-level understanding. (b) A more lightweight architecture by router sharing, applying single LoRA to $W_o, W_{down}$ and using hydra LoRA. (c) A comparison of the time required for 1k steps of training and the proportion of trainable parameters across models with different lightweight designs.

layers to capture information at varying granularities, rather than uniformly combining both types of information across all layers. Further explanations and examples are provided in Appendix B. **Experiments in Appendix E.5 demonstrate that increasing $\alpha^{(l)}$ with $l$ improves model performance. Under this setup, shallow layers focus more on token-level information, while deeper layers emphasize task-level information**, as shown in Figure 2(a).

Additionally, in Appendix C, we introduce a series of lightweight designs to further reduce the number of learnable parameters and enhance computational efficiency. Figure 2(b) provides a brief overview of these designs. A comparison of their training time and parameter efficiency is shown in Figure 2(c).

### 3.3 ENHANCE CERTAINTY AND MAINTAIN BALANCE OF ROUTING RESULTS

In this section, we will discuss the limitations of soft routing and top-$k$ routing introduced in Appendix A and propose a novel auxiliary function to address these issues. Moreover, we find that combining the auxiliary loss with the task router enhances its ability to differentiate between tasks and allows it to generalize to unseen tasks.

We begin by introducing the concept of entropy (Shannon, 1948), which measures the certainty of a probability distribution. Given the gate values $g$ produced by a router, the entropy of $g$ is calculated as:

$$H(g) = -\sum_{j=1}^{e} g_j \log(g_j), \tag{9}$$

where $e$ is the number of experts. **The maximum entropy $\log e$ indicates maximum routing uncertainty (uniform distribution), while the minimum entropy of 0 represents complete certainty in expert selection (point distribution).** Consider a router performing $N$ routing operations in a batch, resulting in a set of distributions $G = \{g^{(1)}, \ldots, g^{(N)}\}$. **The average distribution $\frac{1}{N}\sum_{i=1}^{N} g^{(i)}$ reflects the balance in routing decisions.** The closer this average is to a uniform distribution, the more balanced the selection of experts, with the entropy approaching $\log e$.

We conduct experiments to assess the certainty and balance of top-$k$ and soft routing (see Section 4.1). In soft routing, the shallow layers of LLMs exhibit near-uniform gate values, with entropy approaching $\log e$, indicating random expert selection and a lack of specialization. In deeper layers, while certain experts are more clearly preferred, the selection becomes imbalanced, with a small subset of experts overused, leading to underutilization of others and reduced model performance. A similar issue is observed in top-$k$ routing (Figure 3(b)). Although the load balancing loss helps mitigate imbalance, it comes at the cost of reduced routing certainty.

To address these issues, we propose a novel auxiliary loss function that not only ensures balanced expert selection but also promotes greater certainty in routing results, thereby encouraging more effective specialization of experts across all layers.

To introduce our auxiliary loss, we first present the Generalized Jensen-Shannon (GJS) divergence (Nielsen & Nock, 2009), an extension of the JS divergence (Lin, 1991), which measures the similarity across multiple probability distributions. In GJS, each distribution is assigned a weight, with the sum of these weights equal to 1. We assign a weight of $\frac{1}{N}$ to each gate values. The GJS divergence for $N$ gate values is computed as:

$$\text{GJS}(G) = H\left(\frac{1}{N}\sum_{i=1}^{N} g^{(i)}\right) - \frac{1}{N}\sum_{i=1}^{N} H(g^{(i)}), \tag{10}$$

where $H(\cdot)$ denotes entropy. Maximizing $H\left(\frac{1}{N}\sum_{i=1}^{N} g^{(i)}\right)$ encourages the average distribution to approximate a uniform distribution, promoting the balancing of experts selection. Minimizing $H(g^{(i)})$ increases certainty by driving each individual distribution toward a point distribution. Thus, the GJS divergence, used as an auxiliary loss, promotes both load balancing and more decisive routing decisions. However, our experiments show that directly optimizing this auxiliary loss can reduce model performance by overly constraining model's flexibility. To mitigate this, we propose the **Constrained GJS (CGJS)** divergence:

$$\text{CGJS}(G) = \min\left(H\left(\frac{1}{N}\sum_{i=1}^{N} g^{(i)}\right), \gamma_b \log e\right) - \max\left(\frac{1}{N}\sum_{i=1}^{N} H(g^{(i)}), \gamma_c \log e\right). \tag{11}$$

Here, $\gamma_b$ and $\gamma_c$ are hyperparameters in $[0, 1]$. $\gamma_b$ controls routing balance, with values closer to 1 promoting more balanced experts selection. $\gamma_c$ regulates routing certainty, with values closer to 0 increasing certainty. Fine-tuning $\gamma_b$ and $\gamma_c$ preserves model flexibility, mitigating performance degradation while maintaining expert specialization and balanced experts selection. The definition of the auxiliary loss is as follows:

$$\mathcal{L}_{\text{aux}} = \frac{\max\left((\gamma_b - \gamma_c)\log e - \text{CGJS}(G), 0\right)}{\log e}. \tag{12}$$

The loss is normalized by dividing by $\log e$, ensuring consistency as the number of experts $e$ changes. We apply the auxiliary loss separately to the gate values generated by each task or token router within each batch. **The auxiliary function essentially performs a clustering-like effect.** When applied to task routers, the routing results for similar tasks are brought closer together in the latent space, while dissimilar tasks are driven further apart. **This clustering-like approach enhances the task router's ability to differentiate tasks in an unsupervised manner and generalizes well to unseen tasks.** In Appendix D, we explain in detail, from the perspective of clustering theory, why our auxiliary loss achieves this effect.

Finally, we only optimize the parameters associated with the experts, router, task encoder, and task embedding to minimize the combined language model loss and auxiliary loss. The loss function is defined as:

$$\mathcal{L} = \mathcal{L}_{\text{LM}} + \lambda \sum_{R \in \mathcal{S}} \mathcal{L}_{\text{aux}}^{(R)}, \tag{13}$$

where $\lambda$ is a hyperparameter that adjusts the weight of the auxiliary loss in the overall optimization process. $\mathcal{S}$ represents the set of task routers and token routers, and $\mathcal{L}_{\text{aux}}^{(R)}$ denotes the auxiliary loss computed for each individual task or token router $R$.

## 4 EXPERIMENTS

We fine-tune our model on **Flan v2** (Chung et al., 2024; Longpre et al., 2023), a dataset designed for instruction fine-tuning across 1,836 tasks such as natural language inference, question answering, translation, and sentiment analysis, among others. Fine-tuning on this diverse multi-task dataset enables the model to acquire general problem-solving capabilities rather than simply fitting to a specific dataset.

**Benchmarks and Metrics.** To evaluate multitask performance, we test on several NLP benchmarks, including **MMLU** (Hendrycks et al., 2020), **MMLU-Pro** (Wang et al., 2024), **ARC-Easy**, **ARC-Challenge** (Clark et al., 2018), **OpenBookQA** (Mihaylov et al., 2018), **SWAG** (Zellers et al., 2018), and **CommonsenseQA** (Talmor et al., 2018). These benchmarks consist of multiple-choice questions, assessing various aspects of the model's natural language understanding. For MMLU and MMLU-Pro, we use **macro accuracy**, which averages accuracy across all tasks, while for the other benchmarks, we use **accuracy** as the evaluation metric. More detailed information about the training data and benchmarks is provided in Appendix E.1.

**Base Model and Baseline.** We utilize **Qwen2 1.5B** (Yang et al., 2024) as our base models. For baseline comparisons, we compare HMoRA with full fine-tuning (**Full FT**), **LoRA** ( $r = 8$ and $r = 64$ ) and methods incorporating mixtures of LoRA experts. These methods include **MoLoRA** (Zadouri et al., 2023), **MixLoRA** (Li et al., 2024a) and **HydraLoRA** (Gao et al., 2024). These models were selected due to their similarity in training setup to HMoRA. In contrast, other methods may require predefined task-specific datasets or pretrained LoRA modules, which differ significantly in setup and assumptions. We provide a brief introduction to the baselines in Appendix E.2.

**Training and Evaluation Setup.** We limit the maximum number of training steps to 10,000, conducting evaluations every 200 steps on the validation sets of all benchmarks. If there is no improvement on the validation set for 10 consecutive evaluations, we will terminate the training early. The best checkpoint, determined by the highest averaged accuracy across all benchmarks, is selected for evaluation on the test set. Each experiment is repeated 5 times, and we report the mean of the evaluation metrics.

## 4.1 ROUTING METHODS COMPARISON

We compare the performance of soft routing and top-$k$ routing, as well as the impact of our auxiliary loss ($\mathcal{L}_{\text{aux}}$) and load balancing loss ($\mathcal{L}_{\text{blc}}$) mentioned in Appendix A.

| RM | $\mathcal{L}_{\text{aux}}$ | $\mathcal{L}_{\text{blc}}$ | MMLU | MMLU-Pro | ARC-C | ARC-E | OpenBook | SWAG | Comm | Avg |
|---|---|---|---|---|---|---|---|---|---|---|
| **Soft** | - | - | **55.16** | 24.81 | 69.07 | **85.44** | 81.33 | 53.91 | 70.05 | 62.83 |
| | yes | - | 54.85 | **26.40** | **70.12** | 85.36 | **81.68** | **56.15** | **70.99** | **63.65** |
| **Top-$k$** | - | - | 54.16 | 25.55 | 69.41 | 85.15 | 81.58 | 53.82 | 70.41 | 62.87 |
| | - | yes | 54.09 | 25.44 | 69.12 | **85.56** | 82.02 | 54.63 | 71.42 | 63.19 |
| | yes | - | **54.79** | **26.01** | **69.67** | 85.48 | **82.36** | **55.76** | **71.99** | **63.72** |

Table 1: Performance comparison for different routing methods (**RM**). We calculate the average accuracy (**Avg**) across seven benchmarks as a measure of the model's capability in multi-task scenarios. The best result for soft or top-$k$ routing on each dataset is highlighted in **bold**.

**Implementation Details.** For the MoRA block, we set $r = 8$ and $e = 8$, applying MoRA to all dense layers of Qwen2, including $W_{\text{q}}, W_{\text{k}}, W_{\text{v}}, W_{\text{o}}, W_{\text{gate}}, W_{\text{up}}$, and $W_{\text{down}}$. We only use token-level routing in this section. For $\mathcal{L}_{\text{blc}}$, we used the recommended $\lambda = 0.01$. For $\mathcal{L}_{\text{aux}}$, we set $\lambda = 0.003$, $\gamma_c = 0.4$, and $\gamma_b = 1$. For top-$k$ routing, we set $k = 2$. More training and implementation details as show in Appendix E.3.

**Main Results.** As shown in Table 1, incorporating $\mathcal{L}_{\text{aux}}$ consistently enhances performance, with both soft routing and top-$k$ routing achieving significant accuracy improvements. To further analyze routing certainty and balance, we visualize the entropy of the gate values across different layers in Figure 3. Figure 3 demonstrates that our auxiliary function enhances routing certainty while maintaining relative balance. Although $\mathcal{L}_{\text{blc}}$ improves routing balance, it reduces routing certainty and does not achieve the same performance gains as $\mathcal{L}_{\text{aux}}$. Notably, top-$k$ routing with $\mathcal{L}_{\text{aux}}$ achieves the best average accuracy. Further experimental results on the hyperparameters of $\mathcal{L}_{\text{aux}}$, along with additional analysis of the balance and certainty of routing results, are provided in Appendix E.4.

## 4.2 BASELINE COMPARISON

**Implementation Details.** We compare HMoRA with other fine-tuning methods, employing top-2 routing and the auxiliary function $\mathcal{L}_{\text{aux}}$, along with the hierarchical hybrid routing mentioned in Appendix B. The hyperparameters for hierarchical hybrid routing are set as follows: $\epsilon = 4$,

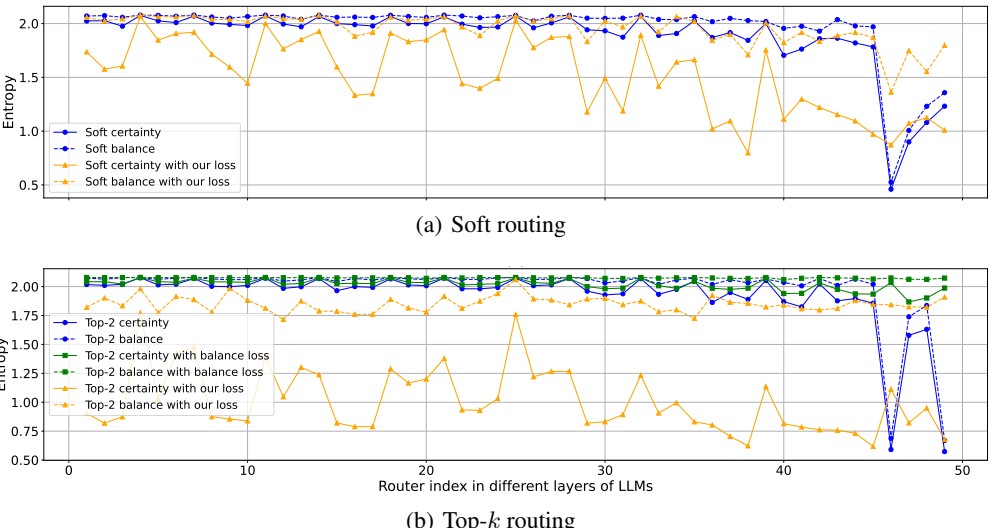

(a) Soft routing

(b) Top-$k$ routing

Figure 3: Visualizing the entropy of the gate values produced by routers across different layers. The base model has 28 layers, but for clarity, we start at layer 4 and sample every 4 layers, resulting in 7 layers. Each layer contains 7 routers (corresponding to $W_q$, $W_k$, $W_v$, $W_o$, $W_{gate}$, $W_{up}$, and $W_{down}$), totaling 49 routers. The dotted lines represent $H\left(\frac{1}{N}\sum_{i=1}^{N} g^{(i)}\right)$, indicating the **balance** of expert selection — **higher values suggest more balanced routing**. The solid lines show $\frac{1}{N}\sum_{i=1}^{N} H(g^{(i)})$, reflecting the **certainty** of routing decisions — **lower values indicate greater certainty**.

| Method | TP | MMLU | MMLU-Pro | ARC-C | ARC-E | OpenBook | SWAG | Comm | Avg |
|---|---|---|---|---|---|---|---|---|---|
| **Full FT** | 100% | 54.12 | 25.09 | 69.13 | 85.10 | 82.35 | 55.11 | 71.16 | 63.15 |
| **LoRA** $r=8$ | 0.60% | 52.89 | 23.77 | 67.52 | 83.92 | 79.96 | 50.51 | 66.72 | 60.76 |
| **LoRA** $r=64$ | 4.78% | 53.34 | 24.24 | 68.60 | 84.90 | 80.93 | 53.33 | 70.64 | 62.28 |
| **MoLoRA** | 3.82% | 53.95 | 25.26 | 69.10 | 85.44 | 81.98 | 54.43 | 70.98 | 63.02 |
| **MixLoRA** | 3.97% | 53.95 | 24.81 | 68.94 | 85.09 | 81.15 | 52.44 | 70.31 | 62.38 |
| **HydraLoRA** | 3.20% | 54.10 | 25.08 | 69.32 | 85.11 | 81.29 | 53.66 | 70.33 | 62.70 |
| **HMoRA** $w$ **LW** | 3.90% | 54.02 | 25.61 | 70.73 | 85.63 | 82.20 | **56.40** | **72.59** | 63.88 |
| **HMoRA** $w/o$ **LW** | 6.31% | **54.63** | **26.59** | **71.47** | **85.87** | **83.23** | 55.28 | 72.08 | **64.16** |

Table 2: Results of baseline comparison experiments across multiple NLP benchmarks. **TP** refers to the percentage of trainable parameters relative to full fine-tuning. The best result for each benchmark is highlighted in **bold**. "$w$ **LW**" and "$w/o$ **LW**" refer to using and not using lightweight designs, respectively.

$\mu = -2$, $\beta_{low} = 0.2$, and $\beta_{high} = 0.8$. All other hyperparameters are consistent with those outlined in Section 4.1. We also compare HMoRA with lightweight designs mentioned in Appendix C, setting $\eta_B = 2$. For LoRA, we conducted experiments with both $r = 8$ and $r = 64$. For other mixtures of LoRA experts models, we fixed $e = 8$ and $r = 8$. Additionally, we performed a hyperparameter search for these baselines and report the best results.

**Main Results.** As shown in Table 2, **HMoRA** $w$ **LW** designs outperforms full fine-tuning on 5 out of 7 benchmarks, while requiring only 3.9% of the trainable parameters. Even on the two benchmarks where **HMoRA** $w$ **LW** slightly lags behind, the performance gap is minimal. **HMoRA** $w/o$ **LW** surpasses full fine-tuning across all benchmarks. Moreover, **HMoRA** $w/o$ **LW** significantly outperforms LoRA and other mixture of LoRA experts methods, demonstrating superior performance across all benchmarks. **HMoRA** $w/o$ **LW** also achieves higher average accuracy. These results demonstrate the effectiveness of HMoRA in efficiently fine-tuning LLMs in a multi-task setting. We also conducted baseline comparison experiments on **LLaMA 3.2 1B**, with the results and analysis provided in Appendix E.7.

## 4.3 ABLATION STUDY

**Ablation Study on the Hyperparameters of the Auxiliary Function.** In Appendix E.4, we conduct ablation experiments primarily on the hyperparameter $\gamma_c$, finding that setting $\gamma_c$ around 0.4 yields better model performance.

**Ablation Study on the Hyperparameters $\epsilon$ and $\mu$ for Hierarchical Hybrid Routing.** In Appendix E.5, we perform ablation experiments on the hyperparameters $\epsilon$ and $\mu$. We find that setting $\epsilon > 0$, i.e., increasing $\alpha^{(l)}$, generally leads to better performance and the model's performance is not sensitive to $\mu$.

**Ablation Study on Lightweight Designs.** In Appendix E.6, we examine the impact of each lightweight design on model performance.

**Quantitative Study on the Ability of Task Routers to Differentiate Unseen Tasks.** We conduct a quantitative study on the performance of task router on unseen tasks in Appendix E.8. The experimental results show that the task router using $\mathcal{L}_{\text{aux}}$ is able to effectively differentiate 42 out of 57 sub-tasks (73.68%) in MMLU. Without any auxiliary function, none of these tasks can be distinguished, while using $\mathcal{L}_{\text{blc}}$ can differentiate just 7 tasks (12.28%).

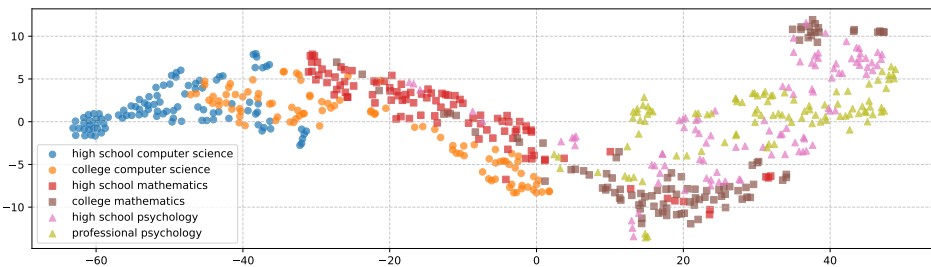

(a) Visualization of gate values from task router with auxiliary loss.

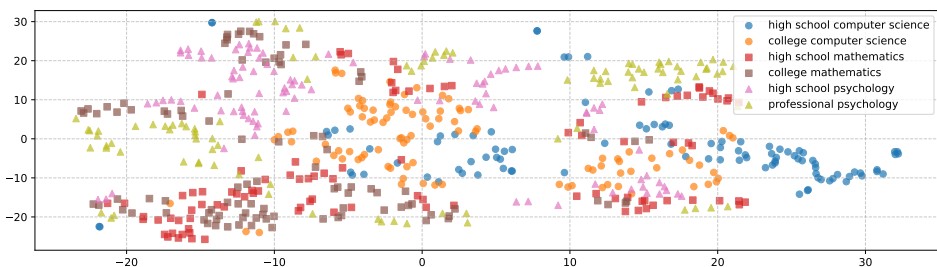

(b) Visualization of gate values from task router without auxiliary loss.

Figure 4: t-SNE visualization of the gate values computed by the final task router in LLMs for different tasks in the MMLU dataset.

**Qualitative Study on the Ability of Task Routers to Differentiate Unseen Tasks.** To more intuitively reveal the impact of the auxiliary loss on the task router, we selected 3 pairs of tasks from MMLU, with each pair containing 2 similar tasks and 100 samples per task. We routed the questions of these tasks and visualized the routing results from the task router in the last laye of LLM with t-SNE (Van der Maaten & Hinton, 2008). As shown in Figure 4(a), when the task router uses $\mathcal{L}_{\text{aux}}$, each task forms a distinct cluster, with similar tasks (represented by the same shape) positioned closer together, while dissimilar tasks are clearly separated. In contrast, the visualization of the routing results from task router without $\mathcal{L}_{\text{aux}}$ is shown in Figure 4(b). The routing results for these tasks, although forming some clusters, are noticeably less distinguishable compared to those using $\mathcal{L}_{\text{aux}}$. This demonstrates that using $\mathcal{L}_{\text{aux}}$ enhances the task router's ability to differentiate tasks, leading to more accurate and distinct routing decisions. Notably, no explicit task labels were provided during training, and the tasks from MMLU were unseen during training. This suggests that the task router learned to differentiate tasks in an unsupervised manner and can generalize to unseen tasks.

| | MMLU | MMLU-Pro | ARC-C | ARC-E | OpenBook | SWAG | Comm | Avg |
|---|---|---|---|---|---|---|---|---|
| HMoRA | **54.63** | **26.59** | **71.47** | **85.87** | **83.23** | **55.28** | **72.08** | **64.16** |
| $w/o\ \mathcal{L}_{\text{aux}}$ **for Task Router** | 53.83 | 25.33 | 70.96 | 85.43 | 82.13 | 53.48 | 71.09 | 63.18 |

Table 3: Ablation study results on the impact of the auxiliary loss for the task router.

As shown in Table 3, when the auxiliary loss is not applied to the task router, the model's performance significantly declines on all benchmarks. This indicates that the auxiliary loss plays a crucial role in the task router's performance.

## 5 RELATED WORK

**Mixture of Experts (MoE).** MoE was introduced by Jacobs et al. (1991) as a framework that divides complex problems into simpler tasks, each handled by a specialized expert. Shazeer et al. (2017) improved the efficiency of this approach by activating only a subset of experts for each input using the Sparsely-Gated MoE layer. Building on this, Lepikhin et al. (2020) scaled MoEs to a 600-billion-parameter multilingual Transformer, enhancing scalability. Du et al. (2022) further scaled this to 1.2 trillion parameters with GLaM, activating a small fraction of the model for each input, resulting in significant computational savings while outperforming GPT-3 on 29 NLP benchmarks. Fedus et al. (2022) simplified the MoE routing algorithm, allowing each token to select one expert, speeding up training without sacrificing quality. Shen et al. (2023) demonstrated that combining MoEs with instruction fine-tuning in LLMs improves performance on task-specific benchmarks while maintaining computational efficiency. Furthermore, MoE is not limited to language modeling but is also extensively utilized in other domains. For instance, Mao et al. (2024) employ MoE to capture travel time uncertainty in road segments under dynamic traffic conditions and Li et al. (2024b) integrates MoE to handle inputs from various modalities.

**Combining MoE with LoRA.** Combining MoE with LoRA enhances LLMs for multi-task learning while significantly reducing the number of trainable parameters. MoLoRA (Zadouri et al., 2023) integrates LoRA with MoEs, achieving performance comparable to full fine-tuning. Luo et al. (2024) employs contrastive learning to promote expert diversity, while Dou et al. (2024) introduces an auxiliary function to specialize experts in either world knowledge or downstream tasks, enhancing overall effectiveness. Xu et al. (2024), Feng et al. (2024), and Zhao et al. (2024) dynamically compose independently trained LoRA experts for different tasks, albeit at the cost of labor-intensive training. Architecturally, MixLoRA (Li et al., 2024a) applies MoE to the feed-forward network and LoRA to self-attention, whereas HydraLoRA (Tian et al., 2025) uses an asymmetric design of LoRA experts to reduce parameters. Gao et al. (2024) demonstrates that assigning more experts to deeper layers improves performance. Task-level routing, as used by Liu et al. (2024) and Feng et al. (2024), assigns tokens based on tasks, improving expert specialization in multi-task scenarios.

## 6 CONCLUSION

This paper introduces HMoRA, an approach that enhances LLMs by integrating a mixture of LoRA experts with hierarchically combined token-level and task-level routing. This design enables the model to capture both fine-grained and global information across different layers of the LLM. We propose a novel auxiliary loss that enhances routing certainty while maintaining a balanced selection of experts, thereby improving expert specialization. It also strengthens task differentiation and enhances generalization to unseen tasks. Additionally, our lightweight designs reduce the parameter size and computational cost, increasing the model's practicality. Experimental results demonstrate that HMoRA outperforms both full-parameter fine-tuning and other LoRA-based approaches across multiple benchmarks. Our auxiliary loss and hybrid routing have the potential to enhance the performance of standard MoE architectures; however, efficiency in industrial-scale applications remains a critical aspect that requires further refinement. This opens promising avenues for future research.

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

## A ROUTING METHODS

The routing methods $f_R(\cdot)$ used in existing MoE methods can be broadly divided into two categories: soft routing and top-$k$ routing. The soft routing is formulated as follows:

$$g = f_R(X) = \text{softmax}(h(X)) \tag{14}$$

Here, $h(\cdot) : \mathbb{R}^{d_{in}} \to \mathbb{R}^e$ represents the routing function of the router. It can be implemented using a simple dense layer or a multi-layer perceptron (MLP). While soft routing is simple, it requires the activation of all experts. This approach is only applied in the mixture of LoRA experts method. In standard MoE models, using soft routing would result in substantial computational overhead. To

address this, standard MoE models typically activate only a few experts sparsely, necessitating the use of top-$k$ routing. The corresponding formula is as follows:

$$g = f_R(X) = \text{softmax}(\text{KeepTopK}(\text{softmax}(h(X)), k)) \tag{15}$$

$$\text{KeepTopK}(v, k)_i = \begin{cases} v_i & v_i \text{ is among the top } k \text{ in } v. \\ -\infty & \text{otherwise.} \end{cases} \tag{16}$$

This approach retains only the top $k$ experts with the highest gate values, setting the rest to zero. Sparse activation is achieved by activating only experts with non-zero gate values. However, top-$k$ routing can easily lead to certain experts being selected frequently, while others are rarely or never selected. This imbalance can degrade the model's generalization ability.

To address this issue, studies such as Shazeer et al. (2017) and Fedus et al. (2022) propose various **load balancing** techniques, which employ auxiliary loss to ensure a more even distribution in the selection of experts, thereby improving the performance of MoE models. The load balancing loss introduced in Fedus et al. (2022) encourages a uniform distribution of tokens across the $e$ experts. The loss function in the original work is applicable to top-1 routing. We extend it to top-$k$ routing. The auxiliary loss for a batch of gate values $G$ is formulated as:

$$\mathcal{L}_{\text{blc}} = e \cdot \sum_{i=1}^{e} F_i \cdot P_i \tag{17}$$

where $F_i$ is the fraction of tokens routed to expert $i$, given by:

$$F_i = \frac{1}{k \cdot |G|} \sum_{g \in G} \mathbf{1}\left[g_i \text{ in top-}k \text{ of } g\right] \tag{18}$$

and $P_i$ is the average router probability for expert $i$, defined as:

$$P_i = \frac{1}{|G|} \sum_{g \in G} g_i \tag{19}$$

This auxiliary loss encourages uniform token routing across experts by minimizing the difference between $F_i$ and $P_i$, ensuring balanced selection of expert.

## B  MAKING HYBRID ROUTING HIERARCHICAL

In Equation 8, there are two hyperparameters, $\epsilon$ and $\mu$. As $l$ increases from 0 to $L$, the value of $\alpha^{(l)}$ transitions from $\sigma(-\epsilon + \mu)$ to $\sigma(\epsilon + \mu)$. If $\epsilon$ is positive, $\alpha^{(l)}$ increases with $l$; conversely, if $\epsilon$ is negative, $\alpha^{(l)}$ decreases as $l$ increases. The larger the value of $\epsilon$, the more rapid this transition. The parameter $\mu$ can be adjusted to control whether more layers employ token routing or task routing. To provide a better understanding, we present examples of various configurations of $\epsilon$ and $\mu$ in Table 4.

| $\epsilon$ | $\mu$ | $\alpha^{(0)}$ | $\alpha^{(1)}$ | $\alpha^{(2)}$ | $\alpha^{(3)}$ | $\alpha^{(4)}$ | $\alpha^{(5)}$ |
|---|---|---|---|---|---|---|---|
| 2 | 0 | 0.12 | 0.23 | 0.40 | 0.60 | 0.77 | 0.88 |
| -2 | 0 | 0.88 | 0.77 | 0.60 | 0.40 | 0.23 | 0.12 |
| 10 | 0 | 0 | 0 | 0.12 | 0.88 | 1 | 1 |
| 10 | 4 | 0 | 0.12 | 0.88 | 1 | 1 | 1 |
| 0 | 0 | 0.5 | 0.5 | 0.5 | 0.5 | 0.5 | 0.5 |

Table 4: Example of different setups of $\alpha^{(l)}$. Here, we set $L = 5$.

Additionally, if $\alpha^{(l)}$ is too small, the task router contributes very little to the final gate values. Conversely, if $\alpha^{(l)}$ is too large, the token router has minimal influence on the final gate values. So we set two thresholds, $\beta_{\text{low}}$ and $\beta_{\text{high}}$. When $\alpha^{(l)} < \beta_{\text{low}}$, we only use token routing, and when $\alpha^{(l)} > \beta_{\text{high}}$, we only use task routing. This allows HMoRA to achieve higher efficiency, as the computational overhead of task routing is significantly lower than that of token routing.

## C  LIGHTER AND MORE EFFECTIVE HMoRA

The inclusion of task encoders and task routers significantly increased the number of trainable parameters. In this section, we explore several designs to reduce the parameter count and computational costs of HMoRA, making it more lightweight and efficient.

**Single LoRA.** As shown in Figure 3, we observed that the mean entropy of the gate values at $W_{\mathrm{o}}$ and $W_{\mathrm{down}}$ was relatively high without $\mathcal{L}_{\mathrm{aux}}$. Even with $\mathcal{L}_{\mathrm{aux}}$, the mean entropy of the gate values at $W_{\mathrm{o}}$ remained higher than the others. This suggests that the inputs to these two dense layers do not demonstrate a clear preference for different experts. As a result, we opted to apply a single LoRA to these dense layers instead of using MoRA.

**Router Sharing.** Whether to assign a router to each dense layer or to a larger module is a question worth considering. Given that the inputs to $W_{\mathrm{q}}$, $W_{\mathrm{k}}$, and $W_{\mathrm{v}}$ are identical, as are those to $W_{\mathrm{up}}$ and $W_{\mathrm{gate}}$, it is reasonable to assign the same router to these dense layers. In contrast, assigning separate routers to each dense layer may increase the complexity of subsequent integration among their outputs. Therefore, we assign a shared router to $W_{\mathrm{q}}$, $W_{\mathrm{k}}$, and $W_{\mathrm{v}}$, and another shared router to $W_{\mathrm{up}}$ and $W_{\mathrm{gate}}$.

**Hydra LoRA+.** Hydra LoRA (Tian et al., 2025) found that the LoRA matrix $W_{\mathrm{A}}$, when trained on different tasks, tends to show similarities across those tasks, with the primary differences manifesting in matrix $W_{\mathrm{B}}$. As a result, matrix $W_{\mathrm{A}}$ can be shared among different experts, capturing task-agnostic knowledge, while matrix $W_{\mathrm{B}}$ specializes in learning domain-specific expertise. Additionally, LoRA+ (Hayou et al., 2024) proposed assigning a higher learning rate (e.g., $\eta_{\mathrm{B}}$ times, where $\eta_{\mathrm{B}} > 1$) to matrix $W_{\mathrm{B}}$ compared to $W_{\mathrm{A}}$, which accelerates convergence and improves performance. In HMoRA, we integrate these approaches.

## D  ENABLING UNSUPERVISED LEARNING THROUGH AUXILIARY LOSS

In this section, we offer an explanation from the perspective of clustering theory, detailing how applying $\mathcal{L}_{\mathrm{aux}}$ enables the task router to distinguish between different tasks in an unsupervised manner and how this capability generalizes to unseen tasks.

From the perspective of clustering theory, the role of our auxiliary function in the task router can be understood as driving the model to automatically discover and distinguish the underlying structures of different tasks. One of the core objectives of clustering theory is to maximize the distance between different clusters (i.e., different tasks), ensuring significant separability between clusters. Another key objective is to minimize the distance or increase the similarity between data points within the same cluster, so that the data points within a cluster are more tightly grouped. Our auxiliary function achieves these goals in the following ways:

- Maximizing $H\left(\frac{1}{N}\sum_{i=1}^{N} g^{(i)}\right)$: The auxiliary function encourages the routing decisions to be distributed more uniformly. This ensures that, on a global scale, routing is not overly concentrated on a few experts but instead utilizes all experts in a balanced manner, ensuring greater differentiation between tasks when selecting experts. This is analogous to the clustering objective of "maximizing inter-cluster distance," where different tasks are assigned to distinct expert clusters.

- Minimizing $\frac{1}{N}\sum_{i=1}^{N} H(g^{(i)})$: The auxiliary function drives the entropy of each individual routing decision toward zero, making the routing decisions more deterministic. This ensures that different inputs from the same task are consistently routed to the same expert, thereby reducing intra-task variability. This is akin to the clustering objective of "minimizing intra-cluster variance," ensuring that all instances of the same task are routed to the same or similar experts.

With this design, the task encoder and task embedding map the task information from the input into a feature space, where similar tasks are positioned closer together, while dissimilar tasks are pushed further apart. The router then utilizes the routing function to assign different task clusters in the feature space to corresponding combinations of experts. This process resembles unsupervised clustering, where the model, by optimizing the auxiliary function, automatically discovers the structural relationships between tasks.

Even when faced with unseen tasks, the task encoder captures the relevant task information from the input and maps it into the feature space. The router then routes the task to the most similar expert cluster based on its position in this space. This capability is analogous to how clustering algorithms assign new data points to the most appropriate cluster by evaluating their proximity to existing clusters.

In summary, the auxiliary function promotes intra-cluster consistency and inter-cluster separability within the framework of clustering theory, enabling the task router to automatically differentiate between tasks in an unsupervised manner. Furthermore, this differentiation capability generalizes to unseen tasks. This mechanism not only enhances the model's performance in multi-task scenarios but also improves its adaptability and robustness in dynamic, evolving environments.

# E  ADDITIONAL EXPERIMENT INFORMATION, RESULTS, AND ANALYSIS

## E.1  DATASET STATISTICS

For training, we utilize the Flan v2 dataset, which is derived from 473 individual datasets, spanning 146 task categories and encompassing a total of 1,836 tasks. The dataset is divided into four mixtures: T0-SF, Muffin, SNI, and CoT. We use a 10-million-sample subset of Flan v2[1] and adjust the proportions of the four mixtures, as shown in Table 5.

|  | T0-SF | Muffin | SNI | CoT |
|---|---|---|---|---|
| **Proportion (%)** | 52.5 | 31.5 | 10.4 | 5.6 |

Table 5: Proportion of different mixtures in Flan v2.

To evaluate multitask performance, we test on several well-established NLP benchmarks:

- **MMLU (Massive Multitask Language Understanding)**: This benchmark covers 57 tasks across various domains, assessing a model's ability to generalize across different subjects.

- **MMLU-Pro**: An extension of MMLU, this variant increases the difficulty by expanding the number of multiple-choice options to ten, making the evaluation more challenging.

- **ARC (AI2 Reasoning Challenge)**: This benchmark tests grade-school science questions, divided into two subsets: **ARC-Easy**, which involves straightforward questions, and **ARC-Challenge**, which requires more complex reasoning and deeper knowledge.

- **OpenBookQA**: A benchmark focused on science questions that require the combination of provided facts with external common knowledge, testing the model's applied reasoning abilities.

- **SWAG (Situations With Adversarial Generations)**: This benchmark evaluates commonsense reasoning by asking the model to predict the most plausible next action in a given scenario.

- **CommonsenseQA**: A multiple-choice dataset that tests commonsense reasoning, requiring models to apply implicit world knowledge to answer questions correctly.

Each of these benchmarks is designed to probe different dimensions of generalization, reasoning, and knowledge application in natural language understanding, providing a comprehensive evaluation of multitask learning performance. Detailed information about these datasets is shown in Table 6.

For each evaluation, testing on full datasets would be highly time-consuming, even exceeding the training time. To address this, we split these benchmarks into smaller subsets. For MMLU, we randomly sample 300 examples from the validation set as the new validation set and 2,000 examples from the test set as the new test set. For MMLU-Pro, we randomly sample 2,000 examples from the test set as the new test set. For SWAG, we randomly sample 300 examples from the original validation set to create a new validation set, and 2,000 examples to form a new test set. For CommonsenseQA, we randomly sample 300 examples from the original validation set to create a new

---

[1]https://huggingface.co/datasets/sordonia/flan-10k-flat

| Dataset | train | validation | test | Number of Options | Metrics |
|---------|-------|------------|------|-------------------|---------|
| **MMLU** | 99842 | 1531 | 14042 | 4 | Macro accuracy |
| **MMLU-Pro** | - | 70 | 12032 | 10 | Macro accuracy |
| **ARC-C** | 1119 | 299 | 1172 | 4 | Accuracy |
| **ARC-E** | 2251 | 570 | 2376 | 4 | Accuracy |
| **OpenBookQA** | 4957 | 500 | 500 | 4 | Accuracy |
| **SWAG** | 73546 | 20006 | 20005 | 5 | Accuracy |
| **CommonsenseQA** | 9741 | 1221 | 1140 | 4 | Accuracy |

Table 6: Information of the original datasets.

| Dataset | validation | test |
|---------|------------|------|
| **MMLU** | 300 | 2000 |
| **MMLU-Pro** | 70 | 2000 |
| **ARC-C** | 299 | 1172 |
| **ARC-E** | 570 | 2376 |
| **OpenBookQA** | 500 | 500 |
| **SWAG** | 300 | 2000 |
| **CommonsenseQA** | 300 | 900 |

Table 7: Number of samples in validation and test splits from the subsets of original datasets.

validation set, and 900 examples to form a new test set. The number of samples in the subsets we created for each benchmark is shown in Table 7. All of our experiments are evaluated on this newly created subset.

### E.2 BRIEF INTRODUCTION OF BASELINE

In this section, we introduce three approaches that combine MoE with LoRA: MoLoRA, MixLoRA, and HydraLoRA. These methods, like our own, do not require datasets with specific task labels or pre-trained LoRA experts, making them suitable for a broad range of tasks. This is the primary reason we selected them as baselines. Below is a brief overview of each method:

- **MoLoRA**: MoLoRA employs token-level soft routing and applies MoRA only to the dense layers within the FFN.
- **MixLoRA**: MixLoRA utilizes token-level top-$k$ routing and applies MoRA to the dense layers within the FFN, while simultaneously fine-tuning each dense layer in the self-attention module using LoRA. Additionally, MixLoRA incorporates a load-balancing loss, $\mathcal{L}_{\text{blc}}$.
- **HydraLoRA**: HydraLoRA introduces a novel structure for LoRA experts, where a shared LoRA matrix $W_A$ is used among all experts, while each expert has its own unique LoRA matrix $W_{B_i}$. This design significantly reduces the number of parameters by sharing part of the LoRA components, which makes the model more parameter-efficient. HydraLoRA employs token-level soft routing.

These methods combine the strengths of MoE and LoRA, making them highly relevant for comparison with our proposed method.

### E.3 TRAINING AND IMPLEMENTATION DETAILS

**General Experimental Setup.** We set the learning rate to $2 \times 10^{-5}$, with a warm-up period of 500 steps during which the learning rate increases linearly. We use the AdamW Loshchilov (2017) optimizer, with a dropout rate of 0.1 and label smoothing set to 0.1. We simulate a base size of 12 using gradient accumulation. The maximum input length is set to 1024, and the maximum output length is set to 512. All the experiments are conducted on NVIDIA A40 GPUs.

**HMoRA Implementation Details.** For the router, we implement it using a single dense layer and apply dropout to its inputs during training. We also apply dropout to the inputs of the experts. For the task encoder, we use a single-layer Transformer encoder, ensuring that the model dimension of task encoder is consistent with the LLM. The number of attention heads is set to 16, and the hidden layer size in the FFN is set to twice the model dimension.

## E.4 ANALYSIS OF THE CERTAINTY AND BALANCE OF DIFFERENT ROUTING METHODS

In this section, we further analyze the impact of the auxiliary loss on the determinism and balance of the routing results. The experimental setup is consistent with that in Section 4.1, except that we adjust the parameter $\gamma_c$ for $\mathcal{L}_{aux}$. We primarily experiment with $\gamma_c$, which affects the certainty of routing results. We fix $\gamma_b = 1$ as we aim for the routing to be as balanced as possible. If the training data itself is imbalanced, setting $\gamma_b$ to a lower value may yield better results. We select $\gamma_c$ values from $\{0, 0.2, 0.4, 0.6, 0.8\}$. The experimental results are shown in Table 8. We observe that for both the top-2 routing and soft routing, setting $\gamma_c$ to 0.4 yields better results, while setting $\gamma_c$ too high or too low negatively affects the performance of the auxiliary function.

| Routing Method | $\gamma_c$ | MMLU | MMLU-Pro | ARC-C | ARC-E | OpenBook | SWAG | Comm | Avg |
|---|---|---|---|---|---|---|---|---|---|
| | 0.0 | 53.87 | 25.00 | 69.18 | 84.90 | 81.21 | **57.14** | 70.59 | 63.13 |
| | 0.2 | 54.47 | 25.57 | 69.12 | 85.00 | 81.14 | 53.44 | 70.25 | 62.71 |
| **Top-2** | 0.4 | **54.79** | **26.01** | **69.67** | **85.48** | 82.36 | 55.76 | **71.99** | **63.72** |
| | 0.6 | 54.59 | 25.13 | 69.49 | 85.44 | **82.76** | 56.14 | 71.39 | 63.56 |
| | 0.8 | 54.39 | 24.25 | 68.89 | 85.25 | 81.08 | 54.41 | 70.70 | 62.71 |
| | 0 | 54.81 | 24.72 | 69.78 | **85.79** | **82.09** | 55.42 | 70.70 | 63.33 |
| | 0.2 | 53.64 | 25.96 | 69.52 | 85.14 | 81.75 | 56.02 | 70.93 | 63.28 |
| **Soft** | 0.4 | **54.85** | **26.40** | 70.12 | 85.36 | 81.68 | **56.15** | 70.99 | **63.65** |
| | 0.6 | 53.19 | 25.04 | 69.09 | 85.17 | 81.41 | 53.29 | 70.16 | 62.48 |
| | 0.8 | 53.94 | 25.23 | 69.78 | 85.59 | 81.82 | 53.56 | **71.16** | 63.01 |

Table 8: Performance comparison of Top-2 and Soft routing methods across various $\gamma_c$ settings on multiple NLP benchmarks.

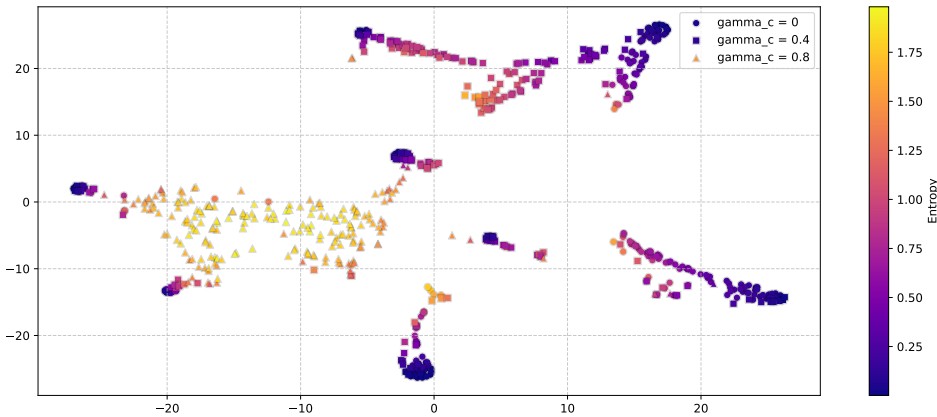

Figure 5: t-SNE visualization of gate values from the top-2 routing with our auxiliary loss at different $\gamma_c$ values. Dark blue points (low entropy) represent high certainty, while yellow points (high entropy) represent low certainty.

Furthermore, to intuitively illustrate the impact of $\gamma_c$ on the certainty of routing, we randomly sampled a data point from the training set and input it into the model. We then visualized the gate values, computed by the final router over the first 200 tokens, using t-SNE. As shown in Figure 5, dark blue points represent gate values with very low entropy, indicating a high level of certainty, as they approach a single-point distribution. In contrast, yellow points represent higher entropy values, indicating lower certainty. The eight dark blue clusters in the figure correspond to eight distinct experts. It can be observed that when $\gamma_c = 0$, most of the gate values are highly concentrated, forming

near single-point distributions. As $\gamma_c$ increases, the certainty of the routing decreases, leading to more dispersed gate values.

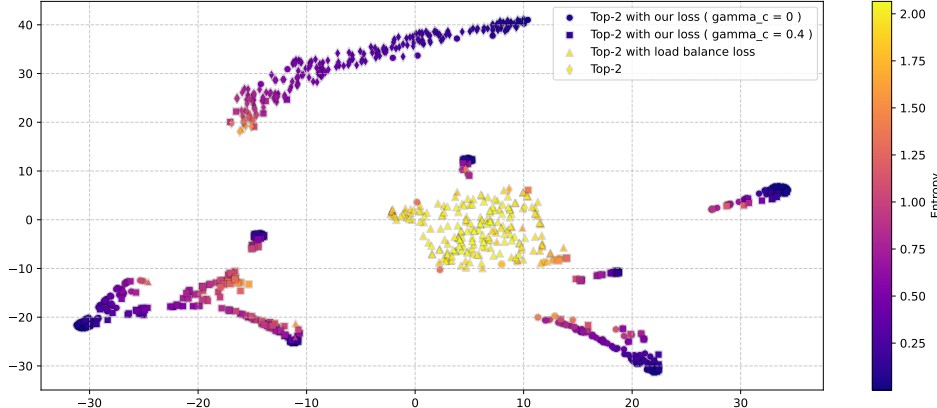

Figure 6: t-SNE visualization of gate values from different routing methods. The plot compares four routing methods: top-2 routing without any auxiliary loss (rhombic), top-2 routing with $\mathcal{L}_{\text{blc}}$ (triangles), and top-2 routing with $\mathcal{L}_{\text{aux}}$ at $\gamma_c = 0$ (circles) and $\gamma_c = 0.4$ (squares). The color scale represents entropy, with darker colors indicating lower entropy (higher certainty) and lighter colors indicating higher entropy (lower certainty).

We compare the certainty of standard top-2 routing, top-2 routing with $\mathcal{L}_{\text{aux}}$, and top-2 routing with $\mathcal{L}_{\text{blc}}$. The results are shown in Figure 6. In top-2 routing without any auxiliary loss, despite the high certainty of the gate values, there is a severe imbalance, with nearly all routing decisions biased toward a single expert (as seen in the upper-middle part of the figure). This causes the MoE model to effectively degrade into a non-MoE model. Conversely, top-2 routing with load balancing loss results in a lack of certainty, with most gate values approaching a uniform distribution, which undermines expert specialization. In contrast, top-2 routing with our auxiliary loss achieves both a relatively balanced selection of experts and greater determinism, thereby enhancing expert specialization.

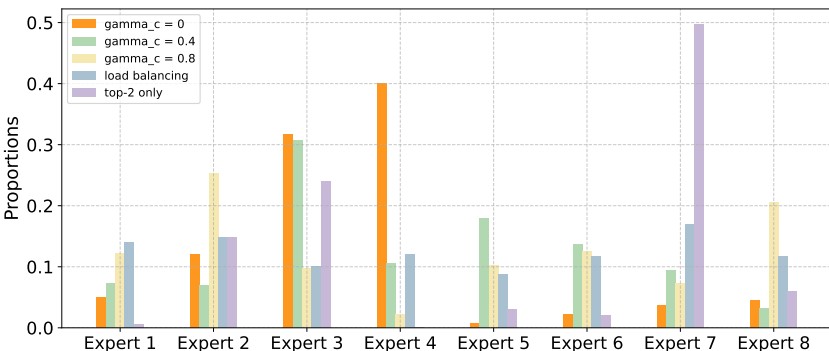

Figure 7: Proportion of activations for each expert in top-2 routing across different configurations. The plot compares the impact of $\mathcal{L}_{\text{aux}}$ with varying $\gamma_c$ values (0, 0.4, and 0.8), and load balancing $\mathcal{L}_{\text{blc}}$ on the selection of experts (1-8).

We further analyze the balance of these routing methods by counting the number of experts activated by the top-2 routing for the first 200 tokens of input and calculating the proportion of activations for each expert relative to the total number of expert activations. Since top-2 routing activates both the expert with the highest probability and the expert with the second-highest probability, the maximum activation proportion for any single expert can reach 50%. The results are shown in Figure 7. Without any auxiliary loss, top-2 routing is the most imbalanced, with Expert 7 being activated nearly 50% of the time, while Expert 1, 4, and 6 are almost never used. A similar imbalance occurs when

$\mathcal{L}_{\text{aux}}$ is applied with $\gamma_c = 0$. However, when $\gamma_c$ is set to 0.4 or 0.8, this imbalance is significantly reduced. The most balanced routing is achieved with top-2 routing using $\mathcal{L}_{\text{blc}}$. However, as noted in the previous analysis, this balance comes at the cost of reduced certainty. Furthermore, given that the data itself is imbalanced, enforcing strict balance can even hinder the specialization of the experts. The experimental results in Section 4.1 also demonstrate that using our auxiliary loss leads to better overall performance.

### E.5 Exploring the Effects of Hybrid Routing

In this section, we investigate how combining task-level and token-level routing can enhance the performance of LLMs. In this set of experiments, we do not apply the lightweight designs. For the auxiliary loss, we set $\gamma_c = 0.4$, $\gamma_b = 1$, and $\lambda = 0.003$. The upper and lower bounds for hybrid routing are set to $\beta_{\text{down}} = 0.2$ and $\beta_{\text{up}} = 0.8$, respectively. We conduct experiments by varying $\epsilon$ and $\mu$.

| Manner | $\epsilon$ | $\mu$ | $\alpha^{(l)}$ | TP | MMLU | MMLU-Pro | ARC-C | ARC-E | OpenBook | SWAG | Comm | Avg |
|---|---|---|---|---|---|---|---|---|---|---|---|---|
| Constant | 0 | -2 | 0 | 5.04% | **54.79** | 26.01 | 69.67 | 85.48 | 82.36 | 55.76 | 71.99 | 63.72 |
| | 0 | -.135 | 0.2 | 6.42% | 54.62 | 25.57 | 69.29 | 84.91 | 81.68 | **56.98** | 70.93 | 63.43 |
| | 0 | -0.4 | 0.4 | 6.42% | 54.76 | 24.72 | 69.81 | 85.22 | 83.03 | 54.41 | 71.39 | 63.34 |
| | 0 | 0.4 | 0.6 | 6.42% | 54.26 | 25.42 | 69.55 | 85.44 | 82.42 | 53.27 | 70.22 | 62.94 |
| | 0 | 1.35 | 0.8 | 6.42% | 53.43 | 25.09 | 69.18 | 85.20 | 82.36 | 54.48 | 71.31 | 63.01 |
| | 0 | 2 | 1 | 6.16% | 53.96 | 26.11 | 69.21 | 85.07 | 82.15 | 54.87 | 70.93 | 63.19 |
| Hierarchical | -4 | 0 | ↓ | 6.29% | 54.30 | 25.47 | 69.55 | 85.69 | 82.42 | 54.78 | 71.05 | 63.32 |
| | 4 | 0 | ↑ | 6.29% | 53.67 | 26.09 | 69.92 | 85.58 | 83.43 | 54.64 | **72.59** | 63.70 |
| | 4 | -2 | ↑ | 6.31% | 54.63 | **26.59** | **71.47** | **85.87** | 83.23 | 55.28 | 72.08 | **64.16** |
| | 4 | 2 | ↑ | 6.26% | 54.50 | 26.14 | 69.89 | 85.66 | 83.42 | 55.89 | 72.23 | 63.96 |

Table 9: The performance on 7 benchmarks when $\alpha^{(l)}$ is set to constant or hierarchical. For the **constant** group of experiments, $\alpha^{(l)}$ is set to a fixed constant across all layers. The value of $\alpha^{(l)}$ shown in the table represents the approximate value under the corresponding settings of $\epsilon$ and $\mu$. For the **hierarchical** group of experiments, ↑ indicates that $\alpha^{(l)}$ increases with layer $l$, while ↓ indicates that it decreases with layer $l$.

As shown in Table 9, we observe that an increasing $\alpha^{(l)}$ across layers significantly boosts the model's performance compared to both decreasing $\alpha^{(l)}$ and constant $\alpha^{(l)}$. The configuration where $\alpha^{(l)}$ increases progressively across layers achieves relatively better average performance, with the setting $\epsilon = 0$ and $\mu = -2$ yielding the best results. This outcome underscores the effectiveness of hierarchical hybrid routing, which enables the model to shift its focus from token-level details in the early layers to broader task-level understanding in the deeper layers. Setting $\epsilon > 0$ (with $\alpha^{(l)}$ increasing) generally yields better performance. The model appears to be less sensitive to the parameter $\mu$.

### E.6 Efficiency Analysis of Lightweight Designs

In this section, we evaluate the lightweight designs discussed in Appendix C. All methods were tested under the same hardware configuration, and we measured the time required to train for 1,000 steps to estimate the computational cost of each lightweight design. For hierarchical hybrid routing, we set $\epsilon = 4$ and $\mu = -2$, with all other settings consistent with those outlined in Appendix E.5. We set the actual batch size to 4, but accumulate gradients over 3 steps to simulate a batch size of 12.

As depicted in Figure 2(c), router sharing does not significantly reduce the number of trainable parameters, but it substantially lowers computational cost. Conversely, both Hydra LoRA and Single LoRA greatly decrease the number of trainable parameters. When these designs are combined, we observe nearly a one-third reduction in both computational cost and trainable parameters.

In terms of performance, as shown in Table 10, router sharing not only maintains performance levels comparable to the base method but even improves the model's performance on the SWAG benchmark. However, both Hydra LoRA and Single LoRA introduce some performance degradation. The combination of Hydra LoRA and LoRA+ helps mitigate the performance drop when $\eta_B = 2$ is applied. Integrating all of the lightweight designs results in only a minimal decrease in performance, demonstrating that these approaches effectively reduce both computational and memory costs with-

| Method | TP | $\eta_B$ | MMLU | MMLU-Pro | ARC-C | ARC-E | OpenBook | SWAG | Comm | Avg |
|---|---|---|---|---|---|---|---|---|---|---|
| Base | 6.31% | - | **54.63** | 26.59 | **71.47** | **85.87** | 83.23 | 55.28 | 72.08 | **64.16** |
| + Router Sharing | 6.22% | - | 54.52 | 25.68 | 71.27 | 85.41 | 82.83 | **56.58** | 72.08 | 64.05 |
| + Single LoRA | 4.77% | - | 53.41 | 25.23 | 69.81 | 85.18 | 81.35 | 56.13 | 70.88 | 63.14 |
| + Hydra LoRA | 4.46% | - | 53.75 | 26.27 | 70.01 | 85.41 | 81.55 | 55.64 | 71.11 | 63.39 |
| + Hydra LoRA+ | 4.46% | 1.2 | 53.78 | 25.39 | 69.35 | 85.41 | 82.82 | 54.30 | 71.36 | 63.20 |
| + Hydra LoRA+ | 4.46% | 1.4 | 53.53 | 25.27 | 69.15 | 85.38 | 81.68 | 54.85 | 71.16 | 63.00 |
| + Hydra LoRA+ | 4.46% | 1.6 | 53.17 | 25.08 | 69.47 | 85.58 | 81.35 | 54.55 | 71.93 | 63.02 |
| + Hydra LoRA+ | 4.46% | 1.8 | 53.79 | 24.99 | 68.89 | 85.52 | **83.50** | 54.78 | 71.65 | 63.30 |
| + Hydra LoRA+ | 4.46% | 2.0 | 53.40 | **26.81** | 70.89 | 85.35 | 82.76 | 55.68 | 71.97 | 63.84 |
| + All | 3.90% | 2.0 | 54.02 | 25.61 | 70.73 | 85.63 | 82.20 | 56.40 | **72.59** | 63.88 |

Table 10: Performance comparison of various lightweight design strategies, including Router Sharing, Single LoRA, Hydra LoRA, and Hydra LoRA+ with different $\eta_B$ settings. The table shows the percentage of trainable parameters (TP), as well as performance across multiple benchmarks. The "+ All" row represents the integration of all lightweight designs.

out significantly compromising the model's accuracy on benchmark tests. These findings highlight the practicality of adopting lightweight designs, especially in resource-constrained environments, as they enable substantial efficiency gains while preserving competitive performance.

### E.7 BASELINE COMPARISON ON LLAMA 3.2 1B

We conducted baseline comparison experiments on LLaMA 3.2 1B. We set $\gamma_c$ to 0.8 and $\eta_B$ to 4. We increased the maximum training steps to 20,000, and evaluated on the validation set every 1,000 steps. All other parameters are consistent with those described in Section 4.2. The experimental results are shown in Table 11.

| | TP | MMLU | MMLU-Pro | ARC-C | ARC-E | OpenBook | SWAG | Comm | Avg |
|---|---|---|---|---|---|---|---|---|---|
| **full fine-tuning** | 100% | 27.42 | 12.05 | 27.51 | 41.22 | 50.91 | 28.47 | 47.75 | 33.61 |
| **LoRA r=8** | 0.45% | 31.4 | 12.21 | 33.42 | 46.63 | 51.82 | 37.62 | 45.81 | 36.99 |
| **LoRA r=64** | 3.64% | 33.82 | 12.11 | **39.12** | 57.57 | 55.15 | 39.44 | 47.59 | 40.68 |
| **MoLoRA** | 2.67% | 25.02 | 9.67 | 26.31 | 26.74 | 27.07 | 25.31 | 22.51 | 23.23 |
| **MixLoRA** | 2.81% | 34.18 | 12.76 | 38.26 | 54.07 | 50.91 | **42.08** | 46.2 | 39.78 |
| **HydraLoRA** | 2.37% | 34.39 | 12.56 | 36.68 | 58.63 | 58.99 | 39.82 | 45.81 | 40.98 |
| **HMoRA $w/o$ LW** | 6.61% | 32.69 | **13.34** | 37.26 | 56.01 | 59.07 | 38.29 | 50.78 | 41.06 |
| **HMoRA $w$ LW** | 4.63% | **35.19** | 12.91 | 38.6 | **58.84** | **61.72** | 39.29 | **54.25** | **42.97** |

Table 11: Baseline comparison based on LLaMA 3.2 1B.

We observed that full fine-tuning resulted in significantly poorer performance compared to PEFT methods. Through our analysis, we summarized two potential reasons for this outcome. First, the Flan dataset does not encompass the knowledge contained in the benchmarks. Consequently, for knowledge-intensive benchmarks such as MMLU and MMLU-Pro, the performance of LLMs on these benchmarks heavily relies on their inherent knowledge. Second, our experiments revealed that LLaMA requires more training steps compared to Qwen to gradually improve its average accuracy across all benchmarks. For full fine-tuning, more fine-tuning steps could lead to greater forgetting of the pre-trained knowledge. In contrast, for PEFT methods, this issue is mitigated as the original model parameters remain frozen during training. Moreover, MoLoRA appeared to be completely ineffective, which we suspect is due to the fact that MoLoRA does not fine-tune the attention layers. Surprisingly, HMoRA $w$ LW achieved the highest average accuracy and performed best on 4 out of 7 benchmarks, even surpassing HMoRA $w/o$ LW.

### E.8 QUANTITATIVE EXPERIMENTS AND ANALYSIS ON TASK ROUTER

In the ablation experiments presented in the paper, we tested 6 unseen tasks from MMLU and used visualizations to verify the task router's ability to differentiate between unseen tasks. Here, we further explore this capability through quantitative analysis. We sampled 100 examples from each of the 57 tasks in MMLU and analyzed the task router's routing results for these samples. First, we

recorded the proportion of expert activations relative to the total number of activations, as shown in Table 12. It is important to note that, since we use top-2 routing, each sample activates two experts (an expert pair). The results show that, without any auxiliary function, all tasks are routed exclusively to experts 1 and 2.

| Expert | 0 | 1 | 2 | 3 | 4 | 5 | 6 | 7 |
|---|---|---|---|---|---|---|---|---|
| with $\mathcal{L}_{aux}$ | 10.21 | 0.45 | 5.21 | 34.11 | 10.67 | 1.72 | 31.25 | 6.34 |
| without any loss | 0 | 0.5 | 0.5 | 0 | 0 | 0 | 0 | 0 |
| with $\mathcal{L}_{blc}$ | 8.47 | 2.11 | 18.37 | 27.87 | 5.62 | 37.55 | 0 | 0 |

Table 12: Proportion of expert activations across different settings, showing the distribution of expert usage with and without auxiliary loss functions.

Further, we analyzed the expert pairs activated for each task. For each task, we identified the most frequently activated expert pair from the 100 samples as the **main expert pair (MEP)**. As show in Table 13, we calculated the proportion of tasks in which the activation of the main expert pair exceeded a certain proportion threshold.

| Threshold | $\geq 0.7$ | $\geq 0.8$ | $\geq 0.9$ | $= 1$ |
|---|---|---|---|---|
| with $\mathcal{L}_{aux}$ | 78.94% | 73.68% | 54.38% | 14.03% |
| without any loss | 100% | 100% | 100% | 100% |
| with $\mathcal{L}_{blc}$ | 21.05% | 12.28% | 3.5% | 1.75% |

Table 13: Proportion of tasks where the main expert pair activation exceeds different thresholds.

**We define tasks with a MEP proportion exceeding 0.8 as recognizable by the router, as this indicates consistent and reliable routing.** Although the proportion of MEP activation consistently reaches 100% without any auxiliary function, it merely routes all tasks to a single pair of experts (1, 2), which we do not regard as a valid indicator of its capability to differentiate between tasks.

| Main Expert Pair | (3, 6) | (2, 7) | (0, 4) | All |
|---|---|---|---|---|
| **All Tasks** | 44 | 5 | 8 | 57 |
| **MEP Proportion $\geq 0.8$ Tasks** | 34 | 3 | 5 | 42 |
| **Ratio** | 77.27% | 60% | 62.5% | 73.68% |

Table 14: Statistics of main expert pairs using $\mathcal{L}_{aux}$, showing the number of tasks for each main expert pair and the proportion of tasks where the main expert pair activation exceeds 0.8.

| Main Expert Pair | (0, 5) | (1, 3) | (2, 3) | (2, 4) | (2, 5) | (3, 5) | All |
|---|---|---|---|---|---|---|---|
| **All Tasks** | 2 | 3 | 1 | 4 | 11 | 36 | 57 |
| **MEP Proportion $\geq 0.8$ Tasks** | 0 | 1 | 0 | 0 | 0 | 6 | 7 |
| **Ratio** | 0% | 33.33% | 0% | 0% | 0% | 16.67% | 12.28% |

Table 15: Statistics of main expert pairs using $\mathcal{L}_{bcl}$, showing the number of tasks for each main expert pair and the proportion of tasks where the main expert pair activation exceeds 0.8.

We further analyzed the MEP for all tasks. The statistics of main expert pairs using $\mathcal{L}_{aux}$ and $\mathcal{L}_{blc}$ are shown in Table 14 and Table 15, respectively. As shown in Table 14, the task router grouped all MMLU tasks into three clusters, with **42 tasks (73.68%) being effectively recognized**. In contrast, when using $\mathcal{L}_{blc}$, although the number of primary expert pairs increased, only two clusters, (1, 3) and (3, 5), were significant, and only 7 tasks (12.28%) were effectively recognized overall. The above quantitative analysis demonstrates that our auxiliary function enhances the task router's ability to differentiate between tasks, even for those that were not encountered during training.

