# OpenReview forum: "HMoRA: Making LLMs More Effective with Hierarchical Mixture of LoRA Experts"
_ICLR.cc/2025/Conference — ICLR 2025 Poster_

### Official Review · Reviewer_wyMT · 2024-11-02

**Soundness:** 2
**Presentation:** 2
**Contribution:** 2
**Rating:** 6
**Confidence:** 3

**Summary:**

In this paper, authors target at the routing problem in existing MoE methods: data granularity used by MoE layer and the generalization to unseen task and expert specialization. The authors introduced HMoRA, a method combining MoE and LoRA used to fine tune LLMs. It includes a hybrid routing strategy that incorporates token-level and task-level routing. They redesign the routing auxiliary loss, and employ lightweight designs to reduce parameters and computational costs.

**Strengths:**

1. The experiments to validate the idea are well designed and thoroughly covers most of the scenarios.

**Weaknesses:**

1. Presentation could be further improved. I feel like the appendix C and D are also important contributions for this idea. However, the organization of the paper make it difficult to identify what is the key contribution and insight of the proposed approach.
2. The idea of mixture of lora and using MoE for both token routing and task routing is not new. I do think a highlight of clear difference between HMoRA and this line of work should be added.

**Questions:**

1. It seems all the components are somehow disconnected. A lot of efforts are put on introducing the new gating algorithm. How is related to apply MoE to LoRa and enable MoE to both token and tasking routing. I feel like the gating algorithm is a separate problem.

---

> ### Author Response · Authors · 2024-11-17
> **Organization of the Paper**
>
> Thank you very much for your thoughtful feedback. We truly appreciate the time and effort you have invested in reviewing our paper. Below, we address each of your concerns in detail and hope to clarify the points you raised:
>
> The main body of the paper focuses on the core structure of the method, which we consider indispensable. Due to the 10-page limit, we had to place many key details in the appendix, despite their importance for understanding certain aspects of our approach. In the upcoming revised version, we will provide a more detailed description of these aspects in the main text to ensure that readers gain a better understanding of our approach.

---

> ### Author Response · Authors · 2024-11-17
> **The Idea of Mixture of Lora and Using MoE for Both Token Routing and Task Routing is Not New**
>
> The key contribution of our work is not merely the combination of MoE and LoRA, but rather the following highlights:
> - The key innovation in our approach is the discovery of **a hierarchical mechanism, where using token-level routing in the shallow layers and gradually transitioning to task-level routing in the deeper layers proves to be more effective**. Althogh PanGu-Σ [1] also combines task-level and token-level routing, first selecting a group of experts based on domain labels, and then randomly routing tokens to one of the experts in that group. This method is limited to predefined domains and cannot generalize to new domains. And it is inefficient because the shallow layers of an LLM primarily capture token-level information, making task-level routing unnecessary at that stage. Our hierarchical mechanism allows the model to capture information at different granularities more effectively. The experiments in Appendix E.5 demonstrate that our hierarchical hybrid routing is more effective than using token-level and task-level routing at every layer. Additionally, J. Zhu et al.[2] proposed various granularity routing methods but did not combine these different granular approaches.
> - The auxiliary function is another key contribution of our work. Our auxiliary function enhances the certainty of routing decisions, resulting in more stable routing and promoting expert specialization. While StableMoE [3] achieves stable routing (in terms of balance and certainty) through a two-stage training and distillation process, we are able to achieve similar stability by simply optimizing our auxiliary function.
> - Additionally, the combination of the auxiliary function and the task router improves the router's ability to differentiate between tasks. While S. Kudugunta  et al.[4] uses sentence representations for routing, thereby eliminating the dependency on task lables, our experiments reveal that directly using sentence representations for routing struggles to effectively differentiate unseen tasks. Our auxiliary function operates in a clustering-like manner, pulling similar tasks closer together in the latent space while pushing dissimilar tasks further apart. This clustering-like approach ensures that the task router’s ability to distinguish between tasks generalizes to unseen tasks as well.
>
> A comparison of our method with other representative MoE approaches is shown in the table below.
>
> | Method               | Stable Routing | Task-level Routing | Token-level Routing | Generalize to Unseen Task | Hierarchical across Layers | Training Complexity           |
> | -------------------- | -------------- | ------------------ | ------------------- | ------------------------- | -------------------------- | ----------------------------- |
> | StableMoE            | Yes            | No                 | Yes                 | -                         | -                          | Complex (two-stage + distill) |
> | PanGu-Σ              | No             | Yes                | Yes                 | No                        | No                         | -                             |
> | Kudugunta et al. [4] | No             | Yes                | Yes                 | No / Weak                 | No                         | -                             |
> | HMoRA                | Yes            | Yes                | Yes                 | Strong                    | Yes                        | Simple (auxiliary loss)       |
>
> We will include a discussion of related work and highlight the strengths of our approach in the revised version of the paper.
>
> [1] X. Ren et al., PanGu-Σ: Towards Trillion Parameter Language Model with Sparse Heterogeneous Computing, arXiv: arXiv:2303.10845, 2023.
>
> [2] J. Zhu et al., Uni-Perceiver-MoE: Learning Sparse Generalist Models with Conditional MoEs, NeurIPS. 2022.
>
> [3] D. Dai et al., StableMoE: Stable Routing Strategy for Mixture of Experts, ACL, 2022.
>
> [4] S. Kudugunta  et al., Beyond Distillation: Task-level Mixture-of-Experts for Efficient Inference, EMNLP 2021

---

> ### Author Response · Authors · 2024-11-17
> **Relation Between LoRA+MoE and other Components**
>
> LoRA+MoE serves as the underlying framework to validate the effectiveness of our **hierarchical hybrid routing** and **auxiliary loss**. Standard MoE models generally require training LLMs from scratch, and verifying our method on such models would incur significant computational costs. In contrast, the LoRA+MoE architecture is commonly used for fine-tuning, which is far more cost-efficient compared to pretraining. Therefore, we initially validated our approach using the LoRA+MoE architecture. However, it is important to emphasize that LoRA+MoE is not the focus of our work. We will clarify this relationship in the revised version of the paper.

---

### Official Review · Reviewer_qmpt · 2024-11-03

**Soundness:** 3
**Presentation:** 3
**Contribution:** 2
**Rating:** 6
**Confidence:** 3

**Summary:**

This paper introduces a hierarchical method, HMoRA, for effective fine-tuning. Specifically, HMoRA integrates token-level and task-level routing to combine MoE and LoRA and utilizes a routing auxiliary loss to improve the certainty of expert selections. Experimental results show that HMoRA outperforms full fine-tuning and some other baseline methods.

**Strengths:**

1. The paper is well-organized and easy to read with clear explanations of the technical details.
2. The approach is straightforward but performs good performance. The hybrid routing that combines token-level and task-level routing is innovative and allows the model to capture multi-granularity information.
3. It is intuitive to apply Generalized Jensen-Shannon divergence for router selection. The experiment results also demonstrate the effectiveness of the auxiliary loss.
4. The overall experimental results show the effectiveness and the potential of the proposed method.

**Weaknesses:**

1. Yet the method shows promising performance in this paper, its novelty is limited in my opinion. The task-level MoE and GJS divergence is not a new topic in MoE or noisy learning areas.
2. The description of task-level routing is somewhat vague. In Figure 2, the inputs include T^in and T^tg. Are these two distinct tasks? How are they set up during training and evaluation?
3. The task representation is encoded based on a task embedding, how can it expand into new tasks? In addition, task routing is a crucial method component, yet there appears to be no ablation experiment focused on task routing within the experiments conducted.

**Questions:**

See Weaknesses.

---

> ### Author Response · Authors · 2024-11-17
> **Novelty of Our Work**
>
> Thank you for your valuable feedback and thoughtful questions. We truly appreciate the time and effort you have invested in reviewing our work. Below, we provide detailed responses to your comments, addressing each point.
>
> - The key innovation in our approach is the discovery of **a hierarchical mechanism, where using token-level routing in the shallow layers and gradually transitioning to task-level routing in the deeper layers proves to be more effective**. Our hierarchical mechanism allows the model to capture information at different granularities more effectively. The experiments in Appendix E.5 also validate the effectiveness of our hierarchical hybrid routing.
> - Although GJS may have applications in other fields, we are the first to apply it to MoE routing. GJS perfectly addresses the issues of imbalance and lack of certainty in routing. Our study provides significant insights and a valuable reference for future MoE research. Furthermore, we propose a new **Constrained GJS divergence (CGJS)** to enhance its effectiveness as a loss function. Directly applying the GJS divergence did not yield satisfactory results, as it severely impacted the router's generalization ability and failed to achieve balanced routing. The experiments in Appendix E.4 further validate the necessity and effectiveness of this adjustment. When $\gamma_c$ is set to 0.4, the model achieves optimal performance. In contrast, when $\gamma_c$ is set to 0, equivalent to optimizing the original GJS, the routing imbalance is as severe as when no auxiliary function is applied.
> - Additionally, in Appendix D, we provide a unique explanation of how the auxiliary function operates similarly to a clustering method, which clarifies why our auxiliary function enhances the task router's routing capabilities. The auxiliary function brings similar tasks closer in the latent space and pushes dissimilar tasks further apart, thereby achieving task differentiation. This clustering-like approach ensures that the task router is capable of generalizing to unseen tasks.

---

> ### Author Response · Authors · 2024-11-17
> **Meaning of  $T^{\text{in}}$ and $T^{\text{tg}}$**
>
> In our paper, $T^{\text{in}}$ and $T^{\text{tg}}$ refer to the input sequence and the output sequence, respectively, as introduced in the Preliminary section. This corresponds to the practical use of LLMs, where $T^{\text{in}}$ represents the input question given to the LLM, and $T^{\text{tg}}$ represents the expected response. This is a common practice in instruction fine-tuning for LLMs. During training, we concatenate $T^{\text{in}}$ and $T^{\text{tg}}$ and feed them together into the LLM, but we only compute the loss and perform backpropagation on the tokens in $T^{\text{tg}}$. Since $T^{\text{tg}}$ is unknown during the LLM's actual deployment, it does not participate in the task routing calculations. Task information is derived solely from $T^{\text{in}}$.

---

> ### Author Response · Authors · 2024-11-17
> **How can task router expand into new tasks?**
>
> - First, since we do not use task labels during training but instead extract semantic information directly from the input for task-level routing, we are not constrained by a predefined set of tasks in the training data, unlike some previous supervised task-level routing methods.
> - Second, the auxiliary function serves a clustering-like role, bringing semantically similar tasks closer in the latent space and pushing dissimilar tasks further apart. This enhances the task router's ability to distinguish between tasks in an unsupervised manner. Similar to clustering methods, when faced with a new task, the router can direct it to the nearest cluster based on its semantics. A detailed explanation of this is provided in Appendix D.

---

> ### Author Response · Authors · 2024-11-17
> **Ablation Experiment Focused on Task Routing**
>
> - Ablation experiments in Section 4.3 demonstrate that the task router can differentiate unseen tasks in MMLU when using the auxiliary function, whereas without it, the router is unable to make such distinctions. This indicates that the auxiliary loss enhances the task router's ability to differentiate between tasks, even for those not encountered during training.
>
> - In Appendix E.5, we conducted ablation studies on the balance of token routing and task routing. The results show that hierarchically combining task-level routing with token-level routing outperforms using either one alone. A constant $\alpha$ indicates that the proportion of task-level routing remains fixed. In contrast, a hierarchical $\alpha$ refers to $\alpha$ either increasing or decreasing across layers. Experimental results show that an increasing $\alpha$ achieves the best performance.The experimental results are as follows:
>
> | Manner           | 𝜖  | 𝜇    | 𝛼  | MMLU      | MMLU-Pro  | ARC-C     | ARC-E     | OpenBook  | SWAG      | Comm      | Avg       |
> | ---------------- | --- | ----- | --- | --------- | --------- | --------- | --------- | --------- | --------- | --------- | --------- |
> | **Constant**     | 0   | -2    | 0   | 54.79     | 26.01     | 69.67     | 85.48     | 82.36     | 55.76     | 71.99     | 63.72     |
> |                  | 0   | -1.35 | 0.2 | 54.62     | 25.57     | 69.29     | 84.91     | 81.68     | **56.98** | 70.93     | 63.43     |
> |                  | 0   | -0.4  | 0.4 | **54.76** | 24.72     | 69.81     | 85.22     | 83.03     | 54.41     | 71.39     | 63.34     |
> |                  | 0   | 0.4   | 0.6 | 54.26     | 25.42     | 69.55     | 85.44     | 82.42     | 53.27     | 70.22     | 62.94     |
> |                  | 0   | 1.35  | 0.8 | 53.43     | 25.09     | 69.18     | 85.44     | 82.36     | 54.48     | 71.31     | 63.01     |
> |                  | 0   | 2     | 1   | 53.96     | 26.11     | 69.21     | 85.07     | 82.15     | 54.87     | 70.93     | 63.19     |
> | **Hierarchical** | -4  | 0     | dec | 54.30     | 25.47     | 69.55     | 85.69     | 82.42     | 54.78     | 71.05     | 63.32     |
> |                  | 4   | 0     | inc | 53.67     | 26.09     | 69.92     | 85.58     | **83.43** | 54.64     | **72.59** | 63.70     |
> |                  | 4   | -2    | inc | 54.63     | **26.59** | **71.47** | **85.87** | 83.23     | 55.28     | 72.08     | **64.16** |
> |                  | 4   | 2     | inc | 54.50     | 26.14     | 69.89     | 85.66     | 83.42     | 55.89     | 72.23     | 63.96     |

---

> ### Comment · Reviewer_qmpt · 2024-11-28
>
> Thank you very much for the response and revision, which addresses some of my concerns.  I will maintain my current rating and advocate acceptance of this paper.

---

### Official Review · Reviewer_nMMJ · 2024-11-03

**Soundness:** 2
**Presentation:** 3
**Contribution:** 3
**Rating:** 6
**Confidence:** 2

**Summary:**

The paper introduces HMoRA (Hierarchical Mixture of LoRA Experts), presenting a significant advancement in fine-tuning LLMs by effectively combining MoE and LoRA with innovative hierarchical routing and auxiliary loss mechanisms. This approach not only improves multi-task performance and expert specialization but also ensures parameter and computational efficiency, making it a promising technique for scalable and versatile natural language processing applications.

**Strengths:**

-  HMoRA successfully combines Mixture of Experts (MoE) with Parameter-Efficient Fine-Tuning (PEFT) methods like LoRA. This integration enables the model to outperform traditional fine-tuning approaches when handling a wide range of tasks. By utilizing specialized experts, HMoRA enhances the model’s ability to generalize across multiple tasks, resulting in outstanding performance on various natural language processing benchmarks.

- One of the standout features of HMoRA is its ability to achieve high performance by fine-tuning only 3.9% of the model’s parameters. This significant reduction in trainable parameters greatly decreases the computational and memory overhead compared to full parameter fine-tuning. Consequently, HMoRA proves to be an exceptionally efficient method for deploying large-scale models in environments with limited resources.

- The paper is well-written, presenting its concepts and methodologies clearly. Additionally, it showcases sufficient innovation, contributing novel ideas and approaches to the field of large language model fine-tuning.

**Weaknesses:**

- The experiments presented in the paper were primarily conducted using the Qwen2 1.5B model and did not extend to larger-scale models (such as those with hundreds of billions of parameters) or to different architectural frameworks. As a result, the scalability and effectiveness of HMoRA on larger or alternative types of LLMs remain unclear.

- Furthermore, although the introduction of lightweight designs is commendable, HMoRA still requires additional computational resources to manage hybrid routing and auxiliary loss functions. However, this increase is understandable given the overall efficiency improvements, and I would not place undue criticism on this aspect.

- Additionally, the auxiliary loss function involves setting multiple hyperparameters (such as γb and γc), which adds complexity to effective tuning across various tasks and datasets. Has there been any investigation into how these hyperparameters impact the ablation studies, and what parameter tuning techniques were employed?

**Questions:**

See weaknesses. Since I am not a researcher in this field, I cannot be certain that the questions I have raised are correct

---

> ### Author Response · Authors · 2024-11-17
> **Experiments on Different Architectural Frameworks**
>
> Thank you for your valuable feedback and thoughtful questions. We truly appreciate the time and effort you have invested in reviewing our work. Below, we provide detailed responses to your comments, addressing each point.
> Since baselines compare  include full fine-tuning, larger models require significantly more resources for fine-tuning. Currently, we lack the resources to conduct experiments on larger models, such as 7B. However, HMoRA is compatible with other architectural models, and we conducted baseline comparison experiments on LLaMA 3.2 1B, and the results are as follows:
>
> |                  | MMLU      | MMLU-Pro  | ARC-C     | ARC-E     | OpenBookQA | SWAG      | Comm      | Avg       |
> | ---------------- | --------- | --------- | --------- | --------- | ---------- | --------- | --------- | --------- |
> | Full Fine-Tuning | 27.42     | 12.05     | 27.51     | 41.22     | 50.91      | 28.47     | 47.75     | 33.61     |
> | LoRA r=8         | 31.4      | 12.21     | 33.42     | 46.63     | 51.82      | 37.62     | 45.81     | 36.99     |
> | LoRA r=64        | 33.82     | 12.11     | **39.12** | 57.57     | 55.15      | 39.44     | 47.59     | 40.68     |
> | MoLoRA           | 25.02     | 9.67      | 26.31     | 26.74     | 27.07      | 25.31     | 22.51     | 23.23     |
> | MixLoRA          | 34.18     | 12.76     | 38.26     | 54.07     | 50.91      | **42.08** | 46.2      | 39.78     |
> | HydraLoRA        | 34.39     | 12.56     | 36.68     | 58.63     | 58.99      | 39.82     | 45.81     | 40.98     |
> | HMoRA            | 32.69     | **13.34** | 37.26     | 56.01     | 59.07      | 38.29     | 50.78     | 41.06     |
> | HMoRA+lw         | **35.19** | 12.91     | 38.6      | **58.84** | **61.72**  | 39.29     | **54.25** | **42.97** |
>
> The experimental results on LLaMA differ somewhat from those on Qwen. Compared to Qwen, LLaMA requires more fine-tuning steps for the average accuracy across multiple benchmarks to gradually converge. With full fine-tuning, an increased number of fine-tuning steps may impair the knowledge learned during pre-training. Additionally, since our fine-tuning data does not include training data specific to the benchmarks, full fine-tuning performs worse than other PEFT methods. Full fine-tuning shows almost no improvement in accuracy on knowledge-intensive benchmarks like MMLU and MMLU-Pro. It only achieves improvements in simpler reasoning tasks such as ARC-E, OpenBookQA, and CommonsenseQA. Furthermore, MoLoRA fine-tuning proved completely ineffective, which we suspect is due to the fact that MoLoRA does not fine-tune the attention layers. Other fine-tuning methods, which freeze the parameters of the initial LLM, preserve the pre-trained knowledge and outperform full fine-tuning. Surprisingly, HMoRA+lw achieved the best average accuracy and performed best on 4 out of 7 benchmarks, even surpassing HMoRA, which has a larger parameter count.

---

> ### Author Response · Authors · 2024-11-17
> **Additional Computational Resources for HMoRA and Ablation Studies of Hyperparameters**
>
> ### Additional Computational Resources for HMoRA
> Thank you for your fairness. We have identified that the task encoder is indeed the primary contributor to the parameter overhead. Optimizing its efficiency will be a key focus of our future work.
>
> ### Ablation Studies of Hyperparameters
>
> We aim to ensure balanced routing, so we fixed $\gamma_b$ at 1. In Appendix E.4, we conducted experiments on $\gamma_c$, selecting values from \{0, 0.2, 0.4, 0.6, 0.8\}. Lower values of $\gamma_c$ result in higher certainty, but this makes achieving balance more difficult, leading to performance degradation. Our experiments showed that a $\gamma_c$ value around 0.4 yields the best results. To further investigate the effect of this hyperparameter, we visualized the distribution of the routing results in the Appendix E.4 and compared it with other auxiliary functions. The experimental results are as follows:
>
>
> | Routing Method | γ_c | MMLU      | MMLU-Pro  | ARC-C     | ARC-E     | OpenBook  | SWAG      | Comm      | Avg       |
> | -------------- | --- | --------- | --------- | --------- | --------- | --------- | --------- | --------- | --------- |
> | **Top-2**      | 0.0 | 53.87     | 25.00     | 69.18     | 84.90     | 81.21     | **57.14** | 70.59     | 63.13     |
> |                | 0.2 | 54.47     | 25.57     | 69.12     | 85.00     | 81.14     | 53.44     | 70.25     | 62.71     |
> |                | 0.4 | **54.79** | **26.01** | **69.67** | **85.48** | 82.36     | 55.76     | **71.99** | **63.72** |
> |                | 0.6 | 54.59     | 25.13     | 69.49     | 85.44     | **82.76** | 56.14     | 71.39     | 63.56     |
> |                | 0.8 | 54.39     | 24.25     | 68.89     | 85.25     | 81.08     | 54.41     | 70.70     | 62.71     |
> | **Soft**       | 0.0 | 54.81     | 24.72     | 69.78     | **85.79** | **82.09** | 55.42     | 70.70     | 63.33     |
> |                | 0.2 | 53.64     | 25.96     | 69.52     | 85.14     | 81.75     | 56.02     | 70.93     | 63.28     |
> |                | 0.4 | **54.85** | **26.40** | **70.12** | 85.36     | 81.68     | **56.15** | 70.99     | **63.65** |
> |                | 0.6 | 53.19     | 25.04     | 69.09     | 85.17     | 81.41     | 53.29     | 70.16     | 62.48     |
> |                | 0.8 | 53.94     | 25.23     | 69.78     | 85.59     | 81.82     | 53.56     | **71.16** | 63.01     |

---

### Official Review · Reviewer_XbfV · 2024-11-04

**Soundness:** 3
**Presentation:** 3
**Contribution:** 3
**Rating:** 6
**Confidence:** 4

**Summary:**

This paper presents HMoRA, a hierarchical fine-tuning method that combines Mixture of Experts (MoE) and LoRA for large language models. The key innovation lies in its hybrid routing mechanism, which hierarchically combines token-level and task-level routing. This allows the model to capture information at different granularities, enhancing its understanding capabilities. Additionally, a novel routing auxiliary loss is introduced to improve the certainty of expert selection and maintain a balanced selection of experts. The method also incorporates several optional lightweight designs that significantly reduce the number of trainable parameters and computational costs without sacrificing much performance.

**Strengths:**

The combination of hierarchical routing and the auxiliary loss function is a novel contribution. The hybrid routing that integrates token-level and task-level routing in a hierarchical manner allows the model to capture different granularities of information, which is a significant improvement over existing methods that focus on only one level of routing.

The experimental results demonstrate that HMoRA outperforms full fine-tuning across multiple NLP benchmarks while fine-tuning only a small percentage (3.9%) of the parameters. This shows the effectiveness of the proposed method in achieving good performance with limited computational resources.

The proposed optional lightweight designs are a practical addition as they significantly reduce both the number of trainable parameters and computational costs without significantly compromising performance. This makes the method more applicable in resource-constrained environments.

**Weaknesses:**

The paper claims that the method can generalize to unseen tasks, but the experiments and analysis related to unseen tasks could be more in-depth. There is a need for more experiments that closely mimic real-world scenarios where the model encounters truly unseen tasks.

While some ablation studies are presented, they could be more comprehensive. For example, for aspects like the hierarchical routing shown in Figure 2(a), more investigation could be done on where exactly to best divide the shallow and deep layers for optimal performance.

**Questions:**

-How well does the method perform on a more diverse set of unseen tasks? Are there any specific characteristics of tasks that might affect its generalization ability? Could the authors provide more details on how the model's performance on unseen tasks was evaluated? What metrics were used and how were the unseen tasks selected?

-Regarding the hierarchical routing in Figure 2(a), what are the criteria for determining the optimal division between shallow and deep layers? How sensitive is the model's performance to this division?

---

> ### Author Response · Authors · 2024-11-17
> **More  Experiments and Analysis Related to Unseen Tasks**
>
> Thank you for your valuable feedback and thoughtful questions. We truly appreciate the time and effort you have invested in reviewing our work. Below, we provide detailed responses to your comments, addressing each point.
>
> It is important to clarify that generalization to unseen tasks refers to the task router's ability to distinguish between different tasks, rather than the inference capabilities of the LLM itself. LLMs inherently possess strong generalization abilities, and our task routing mechanism is designed to enhance the LLM's performance in multi-task scenarios. Therefore, our primary focus is on whether the task router itself can effectively differentiate tasks that were not encountered during training.
>
> In the ablation experiments of the paper, we conducted tests on 6 unseen tasks from MMLU and used visualizations to verify the task router's ability to differentiate between unseen tasks. Furthermore, tests on multiple benchmarks also demonstrate that using the auxiliary function enhances the performance of the LLM.
>
> Here, we delve deeper into this capability through quantitative analysis. We sampled 100 examples from each of all tasks (57) in MMLU and analyzed the task router's routing results for these samples. First, we recorded the number of expert activations, as shown in the table below:
>
> | Expert            | 0     | 1    | 2     | 3     | 4     | 5     | 6     | 7    |
> | :---------------- | :---- | :--- | :---- | :---- | :---- | :---- | :---- | ---- |
> | with aux loss     | 10.21 | 0.45 | 5.21  | 34.11 | 10.67 | 1.72  | 31.25 | 6.34 |
> | without aux loss  | 0     | 0.5  | 0.5   | 0     | 0     | 0     | 0     | 0    |
> | with load balance | 8.47  | 2.11 | 18.37 | 27.87 | 5.62  | 37.55 | 0     | 0    |
>
> It is important to note that, since we use top-2 routing, each sample activates two experts (an expert pair). From the results, we can observe that without the auxiliary function, all tasks are routed to experts 1 and 2.
>
> Further, we analyzed all expert pairs activated for each task. For each task, we identified the most frequently activated expert pair from the 100 samples as the **main expert pair** for that task. We then examined the distribution of main expert pairs across the 57 tasks, calculating the proportion of samples for each task in which the main expert pair was activated, and whether this proportion exceeded a certain threshold. The results are presented in the table below:
>
> |                   | >=0.7  | >=0.8  | >=0.9  | >=1    |
> | :---------------- | :----- | :----- | :----- | :----- |
> | with aux loss     | 78.94% | 73.68% | 54.38% | 14.03% |
> | without aux loss  | 100%   | 100%   | 100%   | 100%   |
> | with load balance | 21.05% | 12.28% | 3.5%   | 1.75%  |
>
> **We define tasks with a main expert pair proportion exceeding 0.8 as recognizable by the router, as this indicates consistent and reliable routing.** Although this proportion is consistently 100% without any auxiliary function, it simply routes all tasks to a pair of experts (1,2), which we do not consider indicative of its ability to differentiate between tasks.
>
> We further analyzed the main expert pairs for all tasks, and the results with the auxiliary function are as follows:
>
> | Main Expert pair | count | >=0.8 | ratio  |
> | :--------------- | :---- | :---- | ------ |
> | (3,6)            | 44    | 34    | 77.27% |
> | (2,7)            | 5     | 3     | 60%    |
> | (0,4)            | 8     | 5     | 62.5%  |
> | All              | 57    | 42    | 73.68% |
>
> As shown in the table, the task router grouped all MMLU tasks into three clusters, with **42 tasks (73.68%) being effectively recognized.**
>
> The results with the load balancing loss are as follows:
>
> | Main Expert pair | count | >=0.8 | ratio  |
> | :--------------- | :---- | :---- | ------ |
> | (0,5)            | 2     | 0     | 0%     |
> | (1,3)            | 3     | 1     | 33.33% |
> | (2,3)            | 1     | 0     | 0%     |
> | (2,4)            | 4     | 0     | 0%     |
> | (2,5)            | 11    | 0     | 0%     |
> | (3,5)            | 36    | 6     | 16.67% |
> | All              | 57    | 7     | 12.28% |
>
> Although the number of primary expert pairs increased, only two clusters, (1,3) and (3,5), were useful, and only 7 tasks (12.28%) were effectively recognized overall.
>
> The above quantitative analysis demonstrates that our auxiliary function enhances the task router's ability to differentiate between tasks, even for those that were not encountered during training.

---

> > ### Author Response · Authors · 2024-11-17
> > **What are the criteria for determining the optimal division between shallow and deep layers? How sensitive is the model's performance to this division?**
> >
> > In fact, we do not strictly categorize the layers as "shallow" or "deep" in a binary manner. As demonstrated in [1], experimental analysis shows that in Transformers, shallow features (e.g., lexical patterns, n-grams) gradually diminish from the early to deeper layers, while semantic features progressively increase. Our approach to balancing token-level and task-level routing follows a similar principle. The coefficient for balancing the two types of routing, $\alpha^l$, is calculated as follows:
> >
> > $$\alpha^l = \sigma\left(-\epsilon + 2 \times \epsilon \times \frac{l}{L} + \mu \right)$$
> >
> > The trend of $\alpha^l$ varying with layer index $l$ is primarily determined by two hyperparameters, $\epsilon$ and $\mu$. When $\epsilon > 0$, the coefficient $\alpha^l$ gradually increases with the layer index $l$. When $\epsilon = 0$, $\alpha^l$ remains the same across all layers. When $\epsilon < 0$, the coefficient $\alpha^l$ decreases as $l$ increases. The smaller the value of $\mu$, the more layers favor task-level routing, whereas larger values of $\mu$ result in more token-level routing. The specific mechanism of this variation is explained in detail with examples in Appendix B.
> >
> > Additionally, in Appendix E.5, we conducted experiments on different configurations of $\alpha^l$.  Our experimental results indicate that **a gradually increasing $\alpha^l$ yields better performance**, while the model is **relatively insensitive to $\mu$.** The following are some of the experimental results, with more analyses provided in Appendix E.5.
> >
> > | Manner           | 𝜖  | 𝜇    | 𝛼^(l) | MMLU  | MMLU-Pro  | ARC-C     | ARC-E     | OpenBook  | SWAG  | Comm      | Avg       |
> > | ---------------- | --- | ----- | ------ | ----- | --------- | --------- | --------- | --------- | ----- | --------- | --------- |
> > | **Constant**     | 0   | -2    | 0      | 54.79 | 26.01     | 69.67     | 85.48     | 82.36     | 55.76 | 71.99     | 63.72     |
> > |                  | 0   | -1.35 | 0.2    | 54.62 | 25.57     | 69.29     | 84.91     | 81.68     | 56.98 | 70.93     | 63.43     |
> > |                  | 0   | -0.4  | 0.4    | 54.76 | 24.72     | 69.81     | 85.22     | 83.03     | 54.41 | 71.39     | 63.34     |
> > |                  | 0   | 0.4   | 0.6    | 54.26 | 25.42     | 69.55     | 85.44     | 82.42     | 53.27 | 70.22     | 62.94     |
> > |                  | 0   | 1.35  | 0.8    | 53.43 | 25.09     | 69.18     | 85.20     | 82.36     | 54.48 | 71.31     | 63.01     |
> > |                  | 0   | 2     | 1      | 53.96 | 26.11     | 69.21     | 85.07     | 82.15     | 54.87 | 70.93     | 63.19     |
> > | **Hierarchical** | -4  | 0     | dec    | 54.30 | 25.47     | 69.55     | 85.69     | 82.42     | 54.78 | 71.05     | 63.32     |
> > |                  | 4   | 0     | inc    | 53.67 | 26.09     | 69.92     | 85.58     | **83.43** | 54.64 | **72.59** | 63.70     |
> > |                  | 4   | -2    | inc    | 54.63 | **26.59** | **71.47** | **85.87** | 83.23     | 55.28 | 72.08     | **64.16** |
> > |                  | 4   | 2     | inc    | 54.50 | 26.14     | 69.89     | 85.66     | 83.42     | 55.89 | 72.23     | 63.96     |
> >
> >
> > [1] M. Geva, R. Schuster, J. Berant, and O. Levy, “Transformer Feed-Forward Layers Are Key-Value Memories,” Sep. 05, 2021, _arXiv_: arXiv:2012.14913. doi: [10.48550/arXiv.2012.14913](https://doi.org/10.48550/arXiv.2012.14913).

---

> ### Comment · Reviewer_XbfV · 2024-12-02
>
> I'm extremely appreciative of the response and the revisions carried out. Given that my assessment score is positive, I shall accordingly retain my current rating and advocate for the acceptance of this paper.

---

### Meta-Review · Area_Chair_2mfc · 2024-12-20

**Metareview:**

- Scientific Claims and Findings:
    - The paper proposes a hierarchical mechanism for finetuning LLMs by combining MoE and LoRA. The key innovation lies in employing token-level routing in the shallow layers and gradually transitioning to task-level routing in the deeper layers, which has proven to be effective.
- Strengths:
   - The proposed method is intuitive and sound.
   - The task routing mechanism is robust to unseen tasks
   - The proposed method outperforms full fine-tuning, LoRA, and methods incorporating mixtures of LoRA experts.
- Weaknesses:
    - The writing, organization, and clarity of the paper could be improved.

- Most Important Reasons for Decision:
     - Based on the identified strengths of the paper.

**Additional Comments On Reviewer Discussion:**

In their rebuttal, the authors emphasized the key novelty of their method and clarified its distinctions from related research. They also detailed the contributions and interconnections of the different components of their approach, effectively addressing the concerns raised by Reviewer wyMT. Additionally, they included further experiments to address other reviewers' concerns and reorganized the content to enhance the clarity of the paper.

Following the rebuttal, Reviewer wyMT increased their rating from 5 to 6, while the other reviewers maintained their ratings of 6. All reviewers are inclined to accept the paper.

---

### Decision · Program_Chairs · 2025-01-22

Accept (Poster)